# Replicability is Asymptotically Free in Multi-armed Bandits

**Junpei Komiyama***
*New York University*
*Mohamed bin Zayed University of Artificial Intelligence (MBZUAI)*
*AIP RIKEN*

**Shinji Ito**
*The University of Tokyo*
*AIP RIKEN*

**Yuichi Yoshida**
*National Institute of Informatics*

**Souta Koshino**
*The University of Tokyo*

**Reviewed on OpenReview:** *https://openreview.net/forum?id=E8rmbq8BYP*

## Abstract

We consider a replicable stochastic multi-armed bandit algorithm that ensures, with high probability, that the algorithm's sequence of actions is not affected by the randomness inherent in the dataset. Replicability allows third parties to reproduce published findings and assists the original researcher in applying standard statistical tests. We observe that existing algorithms require $O(K^2/\rho^2)$ times more regret than nonreplicable algorithms, where $K$ is the number of arms and $\rho$ is the level of nonreplication. However, we demonstrate that this additional cost is unnecessary when the time horizon $T$ is sufficiently large for a given $K, \rho$, provided that the magnitude of the confidence bounds is chosen carefully. Therefore, for a large $T$, our algorithm only requires $K^2/\rho^2$ times smaller amount of exploration than existing algorithms. To ensure the replicability of the proposed algorithms, we incorporate randomness into their decision-making processes. We propose a principled approach to limiting the probability of nonreplication. This approach elucidates the steps that existing research has implicitly followed. Furthermore, we derive the first lower bound for the two-armed replicable bandit problem, which implies the optimality of the proposed algorithms up to a $\log \log T$ factor for the two-armed case.

## 1 Introduction

For scientific findings to be considered valid and reliable, the experimental process must be repeatable and yield consistent results and conclusions across multiple repetitions. A significant issue in many scientific fields is the reproducibility crisis, highlighted by a 2016 survey published in Nature (Baker, 2016), which found that more than 70% of researchers have tried and failed to replicate another researcher's experiments. Similar issues have occasionally been noted in the field of machine learning research. In recent years, major machine learning conferences have started a series of reproducibility programs (e.g., the NeurIPS 2019 Reproducibility Program (Pineau et al., 2021)). Nevertheless, the current trend toward rapid publications in the machine learning field amplifies concerns about reproducibility.

In particular, this paper considers the reproducibility in the domain of sequential learning. We consider the *multi-armed bandit (MAB)* problem (Robbins, 1952; Lai & Robbins, 1985). This problem is one of the

---

*This work was done while the first author was at New York University.

most well-known instances of sequential decision-making problems in uncertain environments. The problem involves conceptual entities called *arms*, of which there are a total of $K$. At each round $t = 1, 2, \ldots$, the forecaster selects one of the $K$ arms and receives a corresponding reward. The forecaster's objective is to maximize the cumulative reward over these rounds. Maximizing this cumulative reward is equivalent to minimizing *regret*, which is the gap between the forecaster's cumulative reward and the reward of the best arm. The initial investigation of this problem took place within the field of statistics (Thompson, 1933; Robbins, 1952). In the past two decades, the research on this problem has been driven by numerous applications, including website optimization (Li et al., 2010), A/B testing (Komiyama et al., 2015) as well as healthcare intervention (Collins et al., 2007).

Several algorithms have proven to be effective. Notably, the upper confidence bound (UCB, Lai & Robbins, 1985; Auer et al., 2002) and Thompson sampling (TS, Thompson, 1933) are widely recognized. Research has shown that these algorithms are asymptotically optimal (Cappé et al., 2013; Agrawal & Goyal, 2012; Kaufmann et al., 2012) in terms of regret, which means that these efficient algorithms exploit accumulated reward information to the fullest extent possible.

## 1.1 Replicability

One possible drawback of such efficiency is the algorithm's lack of reproducibility by other parties, which can make replicating results challenging. This aspect is particularly important because, unlike most statistical learning algorithms, a sequential learning algorithm not only learns from existing data but also actively queries new data. The data-querying process is highly adaptive. For example, a UCB algorithm selects the arm with the largest UCB index, which is calculated based on all previously observed data points. Consequently, even a small modification in the initial data points can lead to a significant change in outcomes. To illustrate this, consider the following example:

**Example 1.** (Crowdsourcing (Abraham et al., 2013; Tran-Thanh et al., 2014)) *Imagine a company conducting a crowd-based A/B testing with $K$ items. In this scenario, each round t corresponds to a worker visiting their website, and each reward represents the feedback provided by the worker, such as a five-star rating. The company runs a bandit algorithm to maximize the total reward.*

It is often the case that the company is reluctant to disclose the dataset due to possible privacy breach (Narayanan & Shmatikov, 2008). Even if a third party can set up an environment that is sufficiently similar to the original, a slight change in the reward of the first few data points can alter subsequent item allocations and significantly affect the performance of a bandit algorithm. In this sense, reproducing sequential algorithms is generally challenging.

It is well-known that the standard frequentist confidence interval no longer holds for the results of the multi-armed bandit problem because such an adaptive algorithm violates the assumption of statistical testing that the number of samples is fixed. Namely, let $N_i$ be the number of samples for each arm $i \in \{1, 2, \ldots, K\}$. Although each run is for fixed duration $T = \sum_i N_i$, the values of $N_i$ can differ across multiple runs. In general, mean statistics derived from the multi-armed bandit algorithm are downward biased (Xu et al., 2013; Shin et al., 2019), and this bias persists even for a large sample scheme (Lai & Wei, 1982), invalidating the use of standard confidence intervals even for asymptotics (Deshpande et al., 2018).

The fundamental notion of reproducibility in this context is *replicability* (Impagliazzo et al., 2022). In simple terms, an algorithm is replicable if a different party with a different dataset is able to replicate the same results.[1] In sequential learning, "the same" refers to maintaining identical sample allocations. Replication of results across different datasets enables us to conduct statistical tests. Suppose that we want to apply a statistical test, such as $z$-test to verify that arm 1 is better than arm 2 with a statistic $z = \sqrt{N_1 + N_2}(\hat{\mu}_1 - \hat{\mu}_2)$, where $N_1, N_2$ are the number of samples and $\hat{\mu}_1, \hat{\mu}_2$ are empirical means. The standard confidence bound states that if $|z| > 1.96$ then we can support the alternative hypothesis, which indicates that one arm is significantly better than the other arm with confidence level $p = 0.05$. This analysis is invalid for the data generated by a multi-armed bandit algorithm, since $N_1, N_2$ are random variables that depend on the rewards. However, when we use a replicable bandit algorithm with non-replication probability at most $\rho = p/2$, then

---

[1] https://www.acm.org/publications/policies/artifact-review-and-badging-current

Table 1: Comparison of regret bounds in the $K$-armed and linear bandit problems. Means $\{\mu_i\}_{i \in [K]}$ are sorted in descending order, $\Delta_i = \mu_1 - \mu_i$ is the suboptimality gap of the arm $i$, and $\Delta = \Delta_2$. RSE has multiple bounds, which implies that it has the smallest regret among the bounds. The lower bound is derived for the case with two arms. Here, $\tilde{O}$ omits a polylog factor in $d, K, T$.

| Problem | Esfandiari et al. (2023a) | This work |
|---------|---------------------------|-----------|
| $K$-armed | $O\left(\sum_{i=2}^{K} \frac{1}{\Delta_i} \frac{K^2 \log T}{\rho^2}\right)$ | $O\left(\sum_{i=2}^{K} \frac{\Delta_i}{\Delta^2}\left(\log T + \frac{\log(K(\log T)/\rho)}{\rho^2}\right)\right)$ (REC (Theorem 10) and RSE (Theorem 11)) $O\left(\sum_{i=2}^{K} \frac{1}{\Delta_i}\left(\log T + \frac{K^2 \log(K(\log T)/\rho)}{\rho^2}\right)\right)$ (RSE, Theorem 11) $O\left(\sqrt{KT\left(\log T + \frac{K^2 \log(K(\log T)/\rho)}{\rho^2}\right)}\right)$ (RSE, Theorem 11) $\Omega\left(\frac{1}{\Delta}\left(\frac{1}{\rho^2 \log((\rho\Delta)^{-1})}\right)\right)$ (Lower bound for $K = 2$, Theorem 12) |
| Linear | $\tilde{O}\left(\frac{K^2 \sqrt{dT}}{\rho^2}\right)$ | $O\left(\frac{d}{\Delta^2}\left(\log T + \frac{\log(K(\log T)/\rho)}{\rho^2}\right)\right)$ (RLSE, Theorem 16) $\tilde{O}\left(\frac{dK}{\rho}\sqrt{T}\right)$ (RLSE, Theorem 16) |

we can guarantee that $N_1, N_2$ are identical with probability at least $1 - p/2$, and if $|z| > 2.25$, which corresponds to the significance level of $p/2$, we can support the alternative hypothesis with $p = p/2 + p/2$. Namely, the probability of false finding under null hypothesis $H_0$ is bounded as:

$$\mathbb{P}[|z| > 2.25 \mid H_0] \leq p,$$

where the probability is taken over the randomness of the algorithm and data. Here, the value 2.25 corresponds to the threshold of $z$-test with $p = 0.05/2$.

## 1.2 Main results

The study of replicability in the literature is limited, as reviewed in Section 9. Existing research suggests that achieving replicability often comes at the cost of sample inefficiency; higher levels of replication require a larger number of samples. In the context of multi-armed bandit algorithms, a smaller non-replication probability $\rho$ is considered to result in higher regret. In particular, Esfandiari et al. (2023a) is the sole work that provides algorithms studying replicability under the same setting.[2] The regret of the algorithms therein is

$$O\left(\sum_{i=2}^{K} \frac{1}{\Delta_i} \frac{K^2 \log T}{\rho^2}\right), \tag{1}$$

which is $K^2/\rho^2$ times larger than that of nonreplicable algorithms, such as UCB and TS. Here, $K$ is the number of arms, $T$ is the number of rounds, $\Delta_i$ is the suboptimality gap for arm $i$, and $\rho$ is the probability of nonreplication. Although the additional factor might appear necessary for the cost of replicability, it is somewhat disappointing to pay such a cost for replication. Upon closer examination of the problem, we discovered that the cost for the replicability can be decoupled from the original regret. Namely, we obtain regret of

$$O\Bigg(\underbrace{\sum_{i=2}^{K} \frac{1}{\Delta_i} \log T}_{\text{Regret for nonreplicable bandits}} + \underbrace{\sum_{i=2}^{K} \frac{1}{\Delta_i} \frac{K^2 \log(K(\log T)/\rho)}{\rho^2}}_{\text{Regret for replicability}}\Bigg). \tag{2}$$

---

[2]Zhang et al. (2025) also considered a more stateful version of bandit problem and derived asymptotic no-regret learning.

This equation 2 is encouraging in the following sense. When we fix $K, \rho$ and consider sufficiently large $T$, then the $O(\log T)$ first term dominates the $O(\log \log T)$ second term. If we ignore the second term, we obtain a regret of $O(\sum_{i=2}^{K} \frac{\log T}{\Delta_i})$, which matches the optimal rate of regret in stochastic bandits. Formally,

$$\limsup_{T \to \infty} \frac{\text{Regret of equation 2}}{\text{Regret of equation 1}} \leq \frac{\rho^2}{K^2},$$

and the regret of equation 2 is rate-optimal, meaning that it matches any (possibly non-replicable) algorithm, such as UCB and TS, up to a universal constant factor. This improvement is attributed to how the factors $\log T$ and $K^2/\rho^2$ are decomposed. To elaborate further, the algorithmic contributions of this work are outlined as follows. First, after the problem setup (Section 2), we introduce a principled framework for bounding nonreplication probability, which is often used implicitly in the literature (Section 3). The framework is quite general and can be extended into a more general class of sequential learning problems, such as episodic reinforcement learning. We then introduce the Replicable Explore-then-Commit (REC) algorithm (Section 4), which has a regret bound of

$$O\left( \sum_{i=2}^{K} \frac{\Delta_i}{\Delta^2} \log T + \sum_{i=2}^{K} \frac{\Delta_i}{\Delta^2} \frac{\log(K(\log T)/\rho)}{\rho^2} \right). \tag{3}$$

The advantage of equation 3 is that both terms are $\tilde{O}(K)$, which is better than equation 2 when the suboptimality gaps for each arm are within a constant factor. However, depending on the suboptimality gaps, equation 3 can be arbitrarily worse compared with the existing bound of equation 1. To deal with this issue, we introduce the Replicable Successive Elimination (RSE) algorithm (Section 5), whose regret bound is the minimum of equation 2 and equation 3. Furthermore, based on these bounds, we derive a distribution-independent regret bound for the replicable $K$-armed bandit problem. Moreover, we derive the first lower bound for the replicable $K$-armed bandits (Section 6) that implies the optimality of equation 2 for $K = 2$ up to a logarithmic factor of $K, \rho, \Delta$, and $\log T$. Finally, we consider the linear bandit problem (Section 7), in which each of the $K$ arms is associated with $d < K$ features. We show that a rather straightforward modification of RSE yields an algorithm whose regret scales linearly in $d$. The performances of REC and RSE are supported by simulations (Section 8). A comparison of existing algorithms and our algorithms is summarized in Table 1.

## 2 Problem Setup

We consider the finite-armed stochastic bandit problem with $T$ rounds. At each round $t$, the forecaster who adopts an algorithm selects one of the arms $I_t \in [K] := \{1, 2, 3, \ldots, K\}$ and receives the corresponding reward $r_t$. Each arm $i \in [K]$ has an (unknown) mean parameter $\mu_i \in \mathbb{R}$. Here, $\mu_i \in [a, b]$ for some $a, b \in \mathbb{R}$ and we let $S = b - a$. We assume $S$ to be a constant. The reward at round $t$ is $r_t = \mu_{I_t} + \eta_t$, where $\eta_t$ is a $\sigma$-subgaussian random variable that is independently drawn at each round.[3] The subgaussian assumption is quite general and is not limited to Gaussian random variables. Any bounded random variable is subgaussian, and thus, it is capable of representing binary events (yes or no) and ordered choice (e.g., 5-star rating). For subgaussian random variables, the following inequality holds.

**Lemma 1** (Concentration inequality). *Let $X_1, X_2, \ldots, X_N$ be $N$ independent (zero-mean) $\sigma$-subgaussian random variables, and $\hat{\mu}_N = (1/N) \sum_i X_i$ be the empirical mean. Then, for any $\epsilon > 0$ we have*

$$\mathbb{P}\left[ |\hat{\mu}_N| \geq \epsilon \right] \leq 2 \exp\left( \frac{-\epsilon^2 N}{2\sigma^2} \right). \tag{4}$$

For ease of discussion, we assume that the mean reward of each arm is distinct. In this case, we can assume $\mu_1 > \mu_2 > \cdots > \mu_K$ without loss of generality. Of course, an algorithm cannot exploit this ordering. A

---

[3]A random variable $\eta$ is $\sigma$-subgaussian if $\mathbb{E}[\exp(\lambda\eta)] \leq \exp(\sigma^2\lambda^2/2)$ for any $\lambda$. For example, a zero-mean Gaussian random variable with variance $\sigma^2$ is $\sigma$-subgaussian.

quantity called regret is defined as follows:

$$\text{Reg}(T) := \sum_{t \in [T]} (\mu_1 - \mu_{I_t}) = \sum_{i \in [K]} \Delta_i N_i(T), \tag{5}$$

where $\Delta_i = \mu_1 - \mu_i$ and $N_i(T)$ is the number of draws on arm $i$ during the $T$ rounds. We also denote $\Delta = \min_{i \geq 2} \Delta_i = \Delta_2$. The performance of an algorithm is measured by the expected regret. Before discussing the replicability, we formalize the notion of dataset in a sequential learning problem because the reward $r_t$ in the aforementioned procedure is drawn adaptively upon the choice of the arm $I_t$. The fact that each noise term $\eta_t$ is drawn independently enables us to reformulate the problem as follows:

**Definition 2.** *(Dataset) The process of the multi-armed bandit problem is equivalent to the following: First, draw a matrix $(r_{i,n})_{i \in [K], n \in [T]}$, where $r_{i,n} = \mu_i + \eta_{i,n}$ and $\eta_{i,n}$ is an independent $\sigma$-subgaussian random variable. Second, run a multi-armed bandit problem. Here, $r_t$ is the $(I_t, N_{I_t}(t))$-entry of the matrix. We call this matrix a dataset and denote it as $\mathcal{D}$. We call $(\mu_i)_{i \in [K]}$ a data-generating process or a model.*

Following Esfandiari et al. (2023a), we consider the class of replicable algorithms that, with high probability, gives exactly the same sequence of selected arms for two independent runs.

**Definition 3.** *($\rho$-replicability, Impagliazzo et al. 2022; Esfandiari et al. 2023a) For $\rho \in [0, 1]$, an algorithm is $\rho$-replicable if,*

$$\mathbb{P}_{U, \mathcal{D}^{(1)}, \mathcal{D}^{(2)}} \left[ (I_1^{(1)}, I_2^{(1)}, \ldots, I_T^{(1)}) = (I_1^{(2)}, I_2^{(2)}, \ldots, I_T^{(2)}) \right] \geq 1 - \rho, \tag{6}$$

*where $U$ represents the internal randomness, and $\mathcal{D}^{(1)}, \mathcal{D}^{(2)}$ are the two datasets that are drawn from the same data-generating process $\{\mu_i\}_{i \in [K]}$. Here, $(I_1^{(1)}, I_2^{(1)}, \ldots, I_T^{(1)})$ (resp. $(I_1^{(2)}, I_2^{(2)}, \ldots, I_T^{(2)})$) is the sequence of draws with $U, \mathcal{D}^{(1)}$ (resp. $U, \mathcal{D}^{(2)}$).*

We may consider $U$ as a sequence of uniform random variables defined on the interval $[0, 1]$. This sequence is utilized by the algorithm to control its behavior, ensuring it selects the same sequence of arms for different datasets, denoted as $\mathcal{D}^{(1)}$ and $\mathcal{D}^{(2)}$. The value $\rho$ corresponds to the probability of nonreplication. The smaller $\rho$ is, the more likely the sequence of actions is replicated. By definition, any algorithm is 1-replicable, and no nontrivial algorithm is 0-replicable. We consider $\rho \in (0, 1)$ as an exogenous parameter, and our goal is to minimize the regret subject to the $\rho$-replicability.

## 3 General Bound of the Probability of Nonreplication

It is not very difficult to see that a standard bandit algorithm, such as UCB, lacks replicability. UCB, in each round, compares the UCB index of the arms, and thus, a minor change in the dataset can alter the sequence of draws $I_1, I_2, \ldots, I_T$. Thus, the design of a replicable algorithm must deviate significantly from standard bandit algorithms. This section presents a general framework for bounding nonreplicable probability in the multi-armed bandit problem. We believe that this framework can be applied to many other sequential learning problems. First, a replicable algorithm should limit its flexibility by introducing phases.[4]

**Definition 4.** *A set of phases is a consecutive partition of rounds $[T]$. Namely, phase $p$ is a consecutive subset of $[T]$, and the first round of phase $p+1$ follows the last round of phase $p$, and each round belongs to one of the phases. We define $P$ to be the number of phases.*

The sequence of draws $I_1, I_2, \ldots, I_T$ can only branch at the end of each phase, as formalized in the following definitions.

**Definition 5.** (Randomness) *The randomness $U$ consists of the randomness for each individual phase. Namely, $U = (U_1, U_2, \ldots, U_P)$.*

**Definition 6.** (Good events, decision variables, and decision points) *We call the end of the final round of each phase a decision point, which we denote as $T_p$. For each $p \in [P]$, we consider the history $\mathcal{H}_p$ to be the set of all results up to the final round $T_p$ of phase $p$. Namely,*

$$\mathcal{H}_p = (I_1, r_1, I_2, r_2, \ldots, I_{T_p}, r_{T_p}) \cup (U_1, U_2, \ldots, U_p). \tag{7}$$

---

[4]Limiting the flexibility of the bandit algorithm has been studied in the literature on batched bandits. Section 9 elaborates on the relation between batched bandits and replicable bandits.

*Each phase $p$ is associated with good event $\mathcal{G}_p(\mathcal{H}_p)$, which is a binary function of $\mathcal{H}_p$. Each phase $p$ is associated with a set of decision variables $d_p$. Decision variables take discrete values and are functions of $\mathcal{H}_p$. Moreover, the sequence of draws on the next phase $\{I_{T_p+1}, I_{T_p+2}, \ldots, I_{T_{p+1}}\}$ is uniquely determined by the decision variables $d_1, d_2, \ldots, d_p$.*

Intuitively speaking, the good events correspond to the concentration of statistics with its probability we can bound with concentration inequalities (by Lemma 1). The set of decision variables uniquely determines the sequence of draws. Note that each phase can be associated with more than one decision variable. To obtain intuition, we consider the following example.

**Example 2.** (A replicable elimination algorithm, Alg 2. in Esfandiari et al. (2023a)) *At the end of each phase, the algorithm obtains an empirical estimate of $\mu_i$ for each arm. It tries to eliminate suboptimal arm $i$, and the corresponding decision variable is*

$$d_{p,i} = \mathbf{1}[\max_j \mathrm{LCB}_j(p) \geq \mathrm{UCB}_i(p)], \tag{8}$$

*where $\mathrm{UCB}_i(p), \mathrm{LCB}_i(p)$ are the (randomized) upper/lower confidence bounds of the arm $i$ at phase $p$. The set of decision variables at phase $p$ is $d_p = (d_{p,i})_i$. Here, $\mathbf{1}[\mathcal{E}]$ is 1 if event $\mathcal{E}$ holds or 0 otherwise. Under good events, by randomizing the confidence bounds with $U_p$, it bounds the probability of nonreplication of each decision variable.*

In the following, we define the nonreplication probability for each component.

**Definition 7.** (Probability of bad event) *Let $\mathcal{G} = \bigcap_p \mathcal{G}_p$ be the global good event and $\rho^G = \mathbb{P}_{U,\mathcal{D}}[\mathcal{G}^c]$ be the probability of the complement event $\mathcal{G}^c$.*

**Definition 8.** (Nonreplication probability of a decision variable) *Let $d_{p,i}$ be the $i$-th decision variable at phase $p$. Its nonreplication probability $\rho^{(p,i)}$ is defined as*

$$\rho^{(p,i)} := \mathbb{P}_{U,\mathcal{D}^{(1)},\mathcal{D}^{(2)}}\left[d_{p,i}^{(1)} \neq d_{p,i}^{(2)} \,\middle|\, \bigcap_{p'=1}^{p-1}\left\{d_{p'}^{(1)} = d_{p'}^{(2)}, \mathcal{G}_{p'}^{(1)}, \mathcal{G}_{p'}^{(2)}\right\}, \mathcal{G}_p^{(1)}, \mathcal{G}_p^{(2)}\right],$$

*where we use superscripts $(1)$ and $(2)$ for the corresponding variables on the two datasets $\mathcal{D}^{(1)}, \mathcal{D}^{(2)}$.*

**Theorem 9.** (Replicability of an algorithm) *An algorithm that involves phases (Definition 4) is $\rho$-replicable with*

$$\rho \leq 2\rho^G + \sum_{p,i} \rho^{(p,i)}. \tag{9}$$

In summary, Theorem 9 enables us to decompose the nonreplication probability $\rho$ into the sum of the nonreplication probability due to the global bad event $\mathcal{G}^c$ and decision variables $(\rho^{(p,i)})_{p,i}$.

## 4 An $O(K)$-regret Algorithm for $K$-armed Bandit Problem

This section introduces the Replicable Explore-then-Commit (REC, Algorithm 1), an $O(K)$-regret algorithm for the $K$-armed bandit problem. This algorithm consists of multiple exploration phases and an exploitation period. At phase $p$, if the algorithm is in the exploration period, we draw each arm up to $N_p := \lceil 8a^{2p}\sigma^2 \rceil$ times, where $a > 1$ is an algorithmic parameter. The last round of each phase is a decision point, where the algorithm decides whether it terminates the exploration period or not. For this aim, it utilizes the minimum suboptimality gap estimator $\hat{\Delta}(p) = \max_i \hat{\mu}_i(p) - \max_i^{(2)} \hat{\mu}_i(p)$, where $\max_i^{(2)}$ denotes the second largest element. This $\hat{\Delta}(p)$ is compared with the confidence bound. There are two keys in the confidence bound $(2 + U_p)\mathrm{Conf}^{\max}(p)$. First, it involves a random variable $U_p \sim \mathrm{Unif}(0,1)$. Second, it is defined as the maximum of $\mathrm{Conf}^{\mathrm{reg}}(p)$ and $\mathrm{Conf}^{\mathrm{repr}}(p)$. Intuitively, $\mathrm{Conf}^{\mathrm{repr}}(p)$ ensures the good event for replicability (c.f., Section 3), while $\mathrm{Conf}^{\mathrm{reg}}(p)$ is for small regret. Here, $\epsilon_p := \frac{1}{a^p}$,

$$\mathrm{Conf}^{\mathrm{repr}}(p) := \epsilon_p \sqrt{\frac{\log(18K^2P/\rho)}{\rho^2}}, \tag{10}$$

---

**Algorithm 1:** Replicable Explore-then-Commit (REC)

---

1  **for** $p = 1, 2, \ldots, P$ **do**
2     Draw shared random variable $U_p \sim \text{Unif}(0, 1)$.
3     Draw each arm for $N_p$ times.
4     **if** $\hat{\Delta}(p) \geq (2 + U_p)\text{Conf}^{\max}(p)$ **then**
5         fix the estimated best arm $\hat{i}^* \in \arg\max_i \hat{\mu}_i(p)$ and break the loop.

6  Draw arm $\hat{i}^*$ for the rest of the rounds.

---

**Algorithm 2:** Replicable Successive Elimination (RSE)

---

1  Initialize the candidate set $\mathcal{A}_1 = [K]$.
2  **for** $p = 1, 2, \ldots, P$ **do**
3     Draw shared random variables $U_{p,i} \sim \text{Unif}(0, 1)$ for $i = 0, 1, 2, \ldots, K$.
4     Draw each arm in $\mathcal{A}_p$ up to $N_p$ times.
5     $\mathcal{A}_{p+1} \leftarrow \mathcal{A}_p$.
6     **if** $\hat{\Delta}(p) \geq (2 + U_{p,0})\text{Conf}^{\max,a}(p)$ **then**
7         $\mathcal{A}_{p+1} = \{\arg\max_i \hat{\mu}_i(p)\}$.       $\rightarrow$ Eliminate all arms except for one.
8     **for** $i \in \mathcal{A}_{p+1}$ **do**
9         **if** $\hat{\Delta}_i(p) \geq (2 + U_{p,i})\text{Conf}^{\max,e}(p)$ **then**
10           $\mathcal{A}_{p+1} \leftarrow \mathcal{A}_{p+1} \setminus \{i\}$.      $\rightarrow$ Eliminate arm $i$.

---

$$\text{Conf}^{\text{reg}}(p) := \epsilon_p \sqrt{\log(KTP)}, \tag{11}$$

and $\text{Conf}^{\max}(p) := \max(C_{\text{mul}}\text{Conf}^{\text{repr}}(p), \text{Conf}^{\text{reg}}(p))$. Here, $C_{\text{mul}} \in \mathbb{R}^+$ is an algorithmic parameter and $P = \min_p\{p : N_p \geq T\} = O(\log T)$ is the maximum number of phases.

The following theorem guarantees the replicability and performance of Algorithm 1.

**Theorem 10.** *Let $C_{\text{mul}} \geq 9/4$ and $a \geq 2$. Assume that $\rho \leq 1/2$. Then, Algorithm 1 is $\rho$-replicable and the following regret bound holds:*

$$\mathbb{E}[\text{Reg}(T)] = O\left(\sum_{i=2}^{K} \frac{\Delta_i}{\Delta^2}\left(\log T + \frac{\log(K(\log T)/\rho)}{\rho^2}\right)\right). \tag{12}$$

## 5  A Generalized Algorithm for $K$-armed Bandit Problem

Although the regret of REC (Algorithm 1) is $O(K)$, which improves upon the existing $O(K^3)$ regret bound in terms of its dependency on $K$, it is not an outright improvement. Considering $K, T, \rho$ as constants, the existing bound given in equation 1 is $O(\sum_i 1/\Delta_i)$. In contrast, REC has a bound of $O(\sum_i \Delta_i/\Delta^2)$. This can be disadvantageous compared to equation 1 in scenarios where the ratio $\Delta_i/\Delta$ is large. To address this issue, we introduce Replicable Successive Elimination (RSE, Algorithm 2). Unlike Algorithm 1, it keeps the list of remaining arms $\mathcal{A}_p$ that it draws. At the end of each phase, it attempts to eliminate all but one arm. If that fails, it attempts to eliminate each arm $i$ individually. Here, let $\hat{\Delta}_i(p) = \max_j \hat{\mu}_j(p) - \hat{\mu}_i(p)$ be the estimated suboptimality gap. For these two elimination mechanisms, we adopt different confidence bounds, which are parameterized by two values $\rho_a, \rho_e > 0$. Let

$$\text{Conf}^{\text{repr},a}(p) := \epsilon_p \sqrt{\frac{\log(18K^2P/\rho_a)}{\rho_a^2}},$$

$$\text{Conf}^{\max,a}(p) := \max(C_{\text{mul}}\text{Conf}^{\text{repr},a}(p), \text{Conf}^{\text{reg}}(p)),$$

$$\text{Conf}^{\text{repr},e}(p) := \epsilon_p \sqrt{\frac{\log(18K^2 P/\rho_e)}{\rho_e^2}},$$

$$\text{Conf}^{\text{max},e}(p) := \max(C_{\text{mul}}\text{Conf}^{\text{repr},e}(p), \text{Conf}^{\text{reg}}(p)),$$

where $\text{Conf}^{\text{reg}}(p)$ is identical to that of Section 4. One can confirm that Algorithm 1 is a specialized version of Algorithm 2 where $(\rho_a, \rho_e) = (\rho, 0)$, where $\rho_e = 0$ implies that the single-arm elimination never occurs. Here, eliminating all but one arm is equivalent to switching to the exploitation period. However, when $\rho_e > 0$, it attempts to eliminate each arm as well. The following theorem guarantees the replicability and the regret of Algorithm 2.

**Theorem 11.** *Let $\rho_a = \rho/2$, and $\rho_e = \rho/(2(K-1))$. Let $C_{\text{mul}} \geq 9/4$ and $a \geq 2$. Assume that $\rho \leq 1/2$. Then, Algorithm 2 is $\rho$-replicable. Moreover, the following three regret bounds hold:*

$$\mathbb{E}[\text{Reg}(T)] = O\left(\sum_{i=2}^{K} \frac{\Delta_i}{\Delta^2}\left(\log T + \frac{\log(K(\log T)/\rho)}{\rho^2}\right)\right),$$

*(same as equation 12)* $\qquad(13)$

$$\mathbb{E}[\text{Reg}(T)] = O\left(\sum_{i=2}^{K} \frac{1}{\Delta_i}\left(\log T + \frac{K^2 \log(K(\log T)/\rho)}{\rho^2}\right)\right), \qquad(14)$$

$$\mathbb{E}[\text{Reg}(T)] = O\left(\sqrt{KT\left(\log T + \frac{K^2 \log(K(\log T)/\rho)}{\rho^2}\right)}\right).$$

*(distribution-independent)* $\qquad(15)$

In particular, the first term of equation 14 matches the optimal regret bound for nonreplicable bandit problem up to a constant factor. Since the second term is $O(\log \log T)$ as a function of $T$, this implies that for any fixed $K, \rho, \{\Delta_i\}$, replicability incurs asymptotically no cost as $T \to \infty$.

## 6 Regret Lower Bound for Replicable Algorithms

The following theorem limits the performance of any replicable bandit algorithm.

**Theorem 12.** *Consider a two-armed bandit problem where rewards are drawn from $\text{Bernoulli}(\mu_i)$ for each arm $i = 1, 2$ with mean parameters $\mu_1, \mu_2$. Consider a $\rho$-replicable algorithm. Then, for any $\Delta > 0$, there exists an instance $(\mu_1, \mu_2)$ with $\Delta = |\mu_1 - \mu_2|$ such that the regret of any $\rho$-replicable bandit algorithm is lower-bounded as*

$$\mathbb{E}[\text{Reg}(T)] = \Omega\left(\min\left(\frac{1}{\rho^2 \Delta \log((\rho\Delta)^{-1})}, T\Delta\right)\right). \qquad(16)$$

**Remark 1.** (Comparison with upper bound) Theorems 10 and 11 state that regret of RSE for the two-armed bandit problem is upper bounded as

$$O\left(\frac{\log T}{\Delta} + \frac{\log((\log T)/\rho)}{\rho^2 \Delta}\right). \qquad(17)$$

The first term corresponds to the regret bound for any uniformly good algorithm.[5] It is well-known that,[6] for a uniformly good algorithm, regret is lower-bounded as $\mathbb{E}[\text{Reg}(T)] = \Omega\left(\frac{\log T}{\Delta}\right)$, which is exactly the first term of equation 17. The second term of equation 17 matches our lower bound $(\Omega(1/\left(\rho^2 \Delta \log((\rho\Delta)^{-1})\right)))$, equation 16) up to a logarithmic factor of $\rho^{-1}, \Delta^{-1}, \log T$. In summary, for $K = 2$, REC and RSE are optimal up to a polylogarithmic factor of $\rho, \Delta$, and $\log T$. In particular, dependence on $T$ is optimal up to a $\log \log T$ factor.

---

[5]An algorithm is uniformly good, if for any $a > 0$ and for any bandit model, it holds that $\mathbb{E}[\text{Reg}(T)] = o(T^a)$ when we view $\Delta$ as a constant.

[6]Theorem 1 in Lai & Robbins (1985).

## 7 An Algorithm for Linear Bandit Problem

Next, we consider the linear bandit problem. In this problem, each arm $i \in [K]$ is associated with a $d$-dimensional feature vector $\boldsymbol{x}_i \in \mathbb{R}^d$ and the reward $r_t$ of choosing an arm $I_t$ is $\boldsymbol{x}_{I_t}^\top \boldsymbol{\theta} + \eta_t$, where $\boldsymbol{\theta}$ is an (unknown) shared parameter vector, and $\eta_t$ is a $\sigma$-subgaussian random variable. Namely, the mean $\mu_i = \boldsymbol{x}_i^\top \boldsymbol{\theta}$ can be estimated via known feature $\boldsymbol{x}_i$ and unknown shared coefficients $\boldsymbol{\theta}$. Without loss of generality, we assume span$(\{\boldsymbol{x}_i\}_{i=1}^K) = \mathbb{R}^d$.

We introduce the replicable linear successive elimination (RLSE). Similarly to RSE (Algorithm 2), this algorithm is elimination-based. The main innovation here is to use the G-optimal design that explores all dimensions in an efficient way. Namely,

**Definition 13.** (G-optimal design) *For $\mathcal{A}_p \subseteq [K]$, let $\pi$ be a distribution over $\mathcal{A}_p$. Let*

$$\boldsymbol{V}(\pi) = \sum_{i \in \mathcal{A}_p} \pi(\boldsymbol{x}_i) \boldsymbol{x}_i \boldsymbol{x}_i^\top, \qquad g(\pi) = \max_{i \in \mathcal{A}_p} ||\boldsymbol{x}_i||^2_{\boldsymbol{V}(\pi)^{-1}}. \tag{18}$$

*A distribution $\pi^*$ is called a G-optimal design if it minimizes $g(\pi)$.*

We use the following well-known result (See, e.g., Section 21 of Lattimore & Szepesvári (2020)).

**Lemma 14** (Kiefer-Wolfowitz). *A G-optimal design $\pi^*$ satisfies $g(\pi^*) = d$. Moreover, there exists a constant approximation of G-optimal design such that $\pi^{*,\mathrm{app}} = \pi^{*,\mathrm{app}}(\mathcal{A}_p)$ with $g(\pi^{*,\mathrm{app}}) \leq 2d$ and its support $\mathrm{Supp}(\pi^{*,\mathrm{app}}) = O(d \log \log d)$.*

An explicit construction of such an approximated G-optimal design is found in the literature[7]. Given an oracle for an approximated G-optimal design, we define the allocation at phase $p$ to be $N_i^{\mathrm{lin}}(p) = \left\lceil N^{\mathrm{lin}}(p) \pi_i^{*,\mathrm{app}} \right\rceil$, where

$$N^{\mathrm{lin}}(p) := 16 a^{2p} \sigma^2 d. \tag{19}$$

Note that $\sum_i N_i^{\mathrm{lin}}(p) \leq N^{\mathrm{lin}}(p) + K$. We use the following lemma for the confidence bound (see e.g., Section 21.1 of Lattimore & Szepesvári 2020):

**Lemma 15** (Fixed-sample bound). *Consider the estimator $\hat{\boldsymbol{\theta}}_p$ at the end of phase $p$. Then, with probability at least $1 - \delta$, the following bound holds uniformly for any $i \in \mathcal{A}_p$:*

$$|\boldsymbol{x}_i^\top (\boldsymbol{\theta} - \hat{\boldsymbol{\theta}}_p)| \leq \frac{\epsilon_p \sqrt{\log(\delta^{-1})}}{2}.$$

*Furthermore, letting $\hat{\mu}_i = \boldsymbol{x}_i^\top \hat{\boldsymbol{\theta}}_p$ and $\hat{\Delta}_{ij} = |\hat{\mu}_i - \hat{\mu}_j|$, we have*

$$|\Delta_{ij} - \hat{\Delta}_{ij}| \leq \epsilon_p \sqrt{\log(\delta^{-1})}. \tag{20}$$

Apart from applying approximated G-optimal exploration, the algorithm closely mirrors the steps of RSE. A comprehensive description of RLSE can be found in Appendix C.

**Theorem 16.** *Let $\rho_a = \rho/2$, and $\rho_e = \rho/(2(K-1))$. Let $C_{\mathrm{mul}} \geq 9/4$ and $a \geq 2$. Assume that $\rho \leq 1/2$. Then, RLSE is $\rho$-replicable. Moreover, the following two regret bounds hold:*

$$\mathbb{E}[\mathrm{Reg}(T)] = O\left(\frac{d}{\Delta^2}\left(\log T + \frac{\log(K(\log T)/\rho)}{\rho^2}\right)\right),$$

*(an $O(d)$ distribution-dependent bound)* $\tag{21}$

$$\mathbb{E}[\mathrm{Reg}(T)] = O\left(d\sqrt{T(\log T \log \log d)\left(\log T + \frac{K^2 \log(K(\log T)/\rho)}{\rho^2}\right)}\right).$$

*(distribution-independent bound)* $\tag{22}$

The first bound depends on $\Delta$ but is $\tilde{O}(1)$ to $K$. The second bound is distribution-independent and is smaller than the existing bound by at least $O(\sqrt{K}/\rho)$ factor (c.f., Table 1). Moreover, for any $K, \rho$, for sufficiently large $T$, the second bound is $O\left(d\sqrt{T(\log T)^2 \log \log d}\right)$.

---

[7]For example, Section 21.2 of Lattimore & Szepesvári 2020.

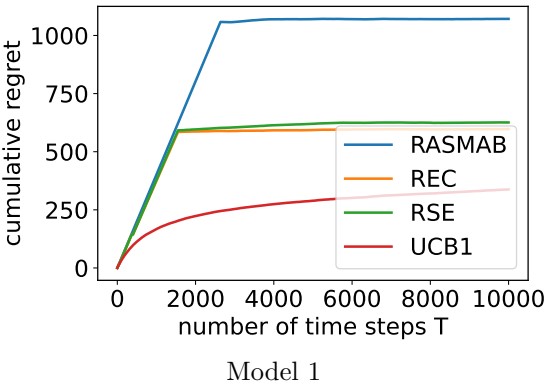 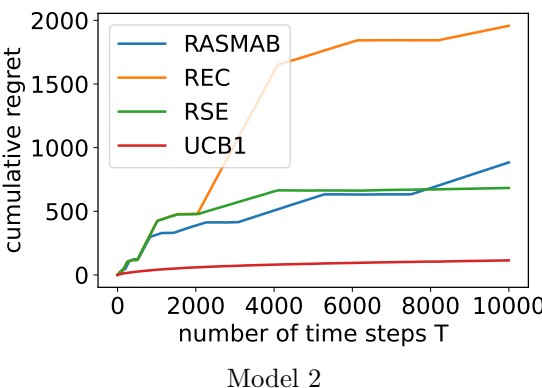

Model 1             Model 2

Figure 1: Regret of algorithms. The horizontal axis indicates the number of rounds $t$ from 1 to $T$, whereas the vertical axis indicates $\text{Reg}(t)$. Results of REC and RSE in Model 1 are very similar.

Table 2: Regret of the algorithms at $T = 10,000$ with two-sigma confidence bounds with a plug-in variance.

|  | Model 1 | Model 2 |
|---|---|---|
| **RASMAB** | $1071.3 \pm 11.0$ | $883.7 \pm 6.2$ |
| **REC (this paper)** | $597.2 \pm 21.1$ | $1956.9 \pm 54.2$ |
| **RSE (this paper)** | $625.8 \pm 28.4$ | $683.1 \pm 13.9$ |
| **UCB1** | $337.7 \pm 6.4$ | $114.4 \pm 6.1$ |

# 8 Simulation

We compared our REC (Algorithm 1) and RSE (Algorithm 2) with RASMAB (Algorithm 2 of Esfandiari et al. (2023a), "Replicable Algorithm for Stochastic Multi-Armed Bandits").[8] Two models of $K$-armed Gaussian bandit problems were considered. To ensure fair comparison, as RASMAB relies on the Hoeffding inequality, we standardized the variance of the arms at 0.5. The results were averaged over 300 runs. We optimize the amount of exploration in REC and RSE, and RASMAB for $\hat{\rho} = 0.3$ by using a grid search.[9] Here, the empirical nonreplication probability $\hat{\rho}$ is obtained by bootstrapping. Namely, assume that the algorithm results in $S$ different sequences of draws, where the corresponding number of occurrences for each sequence are $N_{(1)}, N_{(2)}, N_{(3)}, \ldots, N_{(S)}$. By definition, $\sum_s N_{(s)} = 300$. Then, $\hat{\rho} := 1 - \sum_s (N_{(s)}/300)^2$.

We set the mean parameters as follows: $\boldsymbol{\mu} = (0.7, 0.3, 0.3, 0.3, 0.3, 0.3, 0.3, 0.3, 0.3, 0.3)$ for Model 1 and $\boldsymbol{\mu} = (0.9, 0.8, 0.3)$ for Model 2. The amount of regret is depicted in Figure 1. A lower regret signifies superior performance. Being a nonreplicable algorithm, UCB1 naturally outperforms all other replicable algorithms for both cases.

## 8.1 Discussion on the Results

While the regret analysis of RASMAB is based on a larger confidence bound than that of RSE, the empirical performance of RASMAB under the optimized hyperparameters is similar to RSE where the commitment (i.e., elimination of all arms simultaneously) is never performed. In other words, REC performs simultaneous elimination of all arms (commitment), whereas RASMAB eliminates each arm independently. RSE integrates both elimination mechanisms.

---

[8]We did not include Algorithm 1 of Esfandiari et al. (2023a) because its regret bound is always inferior to RASMAB.

[9]In particular, we multiplied RHS of each decision variable in REC for $C$ times. We multiplied RHS of each commitment and individual elimination decision variable in RSE for $C, C_2$ times. We divided $\beta$ of RASMAB by $C$ times. The values $C, C_2$ of each algorithm were optimized. To make the algorithm comparable, we did not discard sample of previous phases in RASMAB.

In Model 1, commitment is more efficient than individual elimination, and thus REC and RSE outperform RASMAB. In Model 2, individual elimination is more efficient than commitment. As a result, RSE and RASMAB outperform REC.

Table 2 shows the corresponding values of regret at the final round, along with the two-sigma confidence bounds. This implies that all distinctive values in the figure are statistically significant.

## 9 Related Work

Replicability was introduced by Impagliazzo et al. (2022) and they designed replicable algorithms for answering statistical queries, identifying heavy hitters, finding median, and learning halfspaces. Since then, replicable algorithms have been studied for bandit problems (Esfandiari et al., 2023a), reinforcement learning (Eaton et al., 2023; Karbasi et al., 2023), and clustering (Esfandiari et al., 2023b). The equivalence of various stability notions, including replicability and differential privacy (Dwork et al., 2014) was shown for a broad class of statistical problems (Bun et al., 2023). However, the equivalence therein does not necessarily guarantee an efficient conversion. Kalavasis et al. (2023) considered a relaxed notion of replicability. Note also that there are several relevant works Dixon et al. (2023); Chase et al. (2023) that study a different notion of replicability.

**Stability in Sequential Learning**  Stability has also been explored in the context of sequential learning. For example, robustness against corrupted distributions has been examined in the multi-armed bandit problem (Kim & Lim, 2016; Gajane et al., 2018; Kapoor et al., 2019; Basu et al., 2022). Differential privacy has also been considered in this context (Shariff & Sheffet, 2018; Basu et al., 2019; Hu & Hegde, 2022). Differential privacy considers the change of decision against the change of a single data point, whereas in the replicable bandits, we have more than one change of data points between two datasets that are generated from the identical data-generating process. Recent work (Dong & Yoshida, 2023) showed that an algorithm with a low average sensitivity (Varma & Yoshida, 2021) can be transformed to an online learning algorithm with low regret and inconsistency in the random-order setting, and hence in the stochastic setting.

**Batched Bandit Problem**  Prior to the introduction of the replicable bandit algorithm, the batched bandit problem was considered (Auer et al., 2002; Auer & Ortner, 2010; Cesa-Bianchi et al., 2013; Komiyama et al., 2013; Perchet et al., 2016; Garivier et al., 2016; 2019; Gao et al., 2019; Jin et al., 2021a; Esfandiari et al., 2021). In this problem, the algorithm needs to determine the sequence of draws at the beginning of each batch. Existing replicable bandit algorithms in Esfandiari et al. (2023a), as well as our algorithms, adopt phased approaches, and one can find similarities in the algorithmic design. Auer et al. (2002) proposed UCB2, which combines UCB with geometric size of the batches. Regarding the use of explore-then-commit strategy, Perchet et al. (2016) considered the two-armed batched bandit problem. They utilized the fact that the termination of the exploration phases in the explore-then-commit algorithm only occurs in a fixed number of rounds, a concept that we also utilize in the proof of our algorithms. However, their algorithm does not guarantee $\rho$-replicability for $\rho < 1/2$. Our REC extends their results by introducing a randomized confidence level to guarantee a further level of replicability. Furthermore, our RSE generalizes both explore-then-commit and successive elimination (Gao et al., 2019; Esfandiari et al., 2023a) in a replicable way. Given $K, \rho$, for sufficiently large $T$, our algorithm's performance is essentially similar to these batched algorithms Perchet et al. (2016); Gao et al. (2019). Note that, Esfandiari et al. (2023a) also briefly remarked the possibility of the use of the explore-then-commit strategy in replicable bandits. Regarding the optimized performance of explore-then-commit, a slightly adaptive variant can achieve asymptotically optimal regret bounds Jin et al. (2021b) in standard unknown-gap setting Lai & Robbins (1985) as well as in the known-gap Garivier et al. (2016) setting.

## 10 Discussion

We have examined $\rho$-replicable multi-armed bandit algorithms. We demonstrated that the regret of explore-then-commit and elimination algorithms can be expressed as a sum of the standard bandit algorithm's regret and the additional replicability-related regret. For a sufficiently large value of $T$, the latter is negligible.

This represents a significant improvement over existing algorithms. For our analysis, we have developed a framework based on decision variables, which can be used for bounding the probability of nonreplication in larger classes of sequential replicable learning problems, such as reinforcement learning problems.

The notion of replicability in this paper is defined to be the probability of two independent runs of an algorithm resulting in the same sequence of decisions. The definition makes sense when we consider the adaptive clinical trials as well as educational applications, where we want to ensure the treatment assignment of the same patient or the same student is the same across different runs. However, this is a strong notion of replicability, and it is an interesting future direction to consider a relaxed notion of replicability. For example, we can consider a notion of replicability where two independent runs of an algorithm result in the same number of draws with high probability. A more relaxed notion of replicability can be defined as the one that only guarantees the same number of draws of the top arm. Such relaxed replicability is meaningful in conducting statistical testing problems.

Even though we have shown that the leading $\log T$ term is free of non-replicability parameter $\rho$, the $(\Delta\rho)^{-2}$ term, which is larger than $\Delta^{-2}$ of standard bandit algorithms by $\rho^{-2}$, still appears in the sublogarithmic term and it is necessary to have such a term as we show in the lower bound proof. An interesting future direction is to consider whether we can remove the $(\Delta\rho)^{-2}$ term in a relaxed setting. The term is derived from the necessity of estimating the gap $\Delta_i$ with precision of $\rho \times \Delta_i$, which is required for guaranteeing $\rho$-replicability. However, if we assume the gap $\Delta_i$ is known (Garivier et al., 2016), we hypothesize that it is possible to remove the $(\Delta\rho)^{-2}$ term. In this case, the double explore-then-commit algorithm (Jin et al., 2021b) might be adapted to the replicable setting, and it can achieve the optimal regret bound without the $(\Delta\rho)^{-2}$ term.

### 10.1 Summary for Practitioners

Replicability guarantees that the decison of algorithm depends on the initial random seed but is independent of the randomness of the data. Popular sequential learning algorithms, such as UCB and Thompson sampling, are non-replicable. Explore-then-commit algorithms and batched elimination algorithms are replicable by choosing an appropriate threshold. This paper showed that the cost of replicability is not multiplicative in terms of $T$ but is additive. That is, for sufficiently large $T$, we can design batched algorithms that are replicable and have the same leading term of regret as non-replicable algorithms, up to a constant factor. Roughly speaking, if

$$\log T \geq \frac{D^2 \log(K/\rho)}{\rho^2},$$

then the replicability cost is negligible, where $D$ is the number of decisions in the algorithm, which is 1 for explore-then-commit algorithms and $K$ for elimination algorithms.

## Acknowledgement

We thank the anonymous reviewers for their helpful comments. These comments helped us to improve the accuracy of the paper, add discussion on the known gap case, correct the errors in the proofs, and add simulations on linear bandits. S.I. is supported by JSPS KAKENHI Grant Number JP25K03184 and by JST PRESTO, Japan, Grant Number JPMJPR2511. Y.Y. is supported by JSPS KAKENHI Grant Number JP24K02903.

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

Table 3: Notation in this paper.

| symbol | definition |
|---|---|
| $K$ | number of the arms |
| $T$ | number of the time steps |
| $\mu_i$ | mean reward of arm $i \in [K]$, which we assume $\mu_1 > \mu_2 > \cdots > \mu_K$ |
| $\rho$ | level of non-replication |
| $\Delta_i$ | suboptimality gap $(= \mu_1 - \mu_i)$ |
| $\Delta$ | minimum suboptimality gap $(= \mu_1 - \mu_2)$ |
| $I_t$ | the arm drawn at time step $t$ |
| $r_t$ | reward at time step $t$ |
| $\sigma$ | subgaussian radius |
| $\eta_t$ | an i.i.d. noise on the reward at time step $t$, which is $\sigma$-subgaussian |
| $N_i(t)$ | number of draws on arm $i$ for the first $t$ rounds |
| $\mathrm{Reg}(T)$ | regret (performance measure, equation 5) |
| $U$ | randomness. Given the history, an algorithm's behavior is determined by $U$. |
| $\mathbf{1}[\mathcal{X}]$ | $= 1$ if $\mathcal{X}$ holds or $= 0$ otherwise |
| $P$ | number of phases for a phased algorithm (Definition 4). |
| $U_p$ | randomness for phase $p$ for a phased algorithm (Definition 5). |
| $d_{p,i}$ | $i$-th decision variable at phase $p$ (Definition 6) |
| $\mathcal{G}_p$ | good event at phase $p$ (Definition 6) |
| $\rho^G$ | nonreplication probability due to global bad event $\mathcal{G}^c$ (Definition 7) |
| $\rho^{(p,i)}$ | nonreplication probability due to $d_{p,i}$ (Definition 8) |
| $a$ | algorithmic parameter $(\geq 2)$ (Section 4) |
| $N_p$ | $= \lceil 8a^{2p}\sigma^2 \rceil$ (Section 4) |
| $\hat{\mu}_i(p), \hat{\Delta}(p), \hat{\Delta}_i(p)$ | corresponding empirical analogues at phase $p$ (Sections 4 and 5) |
| $C_{\mathrm{mul}}$ | algorithmic parameter $(\geq 9/4)$ (Section 4) |
| $\epsilon_p$ | $= a^{-p}$ (Section 4) |
| $\mathrm{Conf}^{\mathrm{repr}}(p)$ | confidence bound used by REC (Section 4) |
| $\mathrm{Conf}^{\mathrm{reg}}(p)$ | confidence bound used by all algorithms (Section 4) |
| $\mathrm{Conf}^{\mathrm{max}}(p)$ | $= \max(C_{\mathrm{mul}}\mathrm{Conf}^{\mathrm{repr}}(p), \mathrm{Conf}^{\mathrm{reg}}(p))$ (Section 4) |
| $\mathcal{A}_p \subset [K]$ | remaining arms at phase $p$ (Section 5) |
| $U_{p,i}$ | randomness at phase $p$, where $U_p = (U_{p,0}, U_{p,1}, \ldots, U_{p,K})$ (Section 5) |
| $\mathrm{Conf}^{\mathrm{repr},a}(p)$ | confidence bound used by RSE and RLSE (Section 5) |
| $\mathrm{Conf}^{\mathrm{repr},e}(p)$ | confidence bound used by RSE and RLSE (Section 5) |
| $\mathrm{Conf}^{\mathrm{max},a}(p)$ | $= \max(C_{\mathrm{mul}}\mathrm{Conf}^{\mathrm{repr},a}(p), \mathrm{Conf}^{\mathrm{reg}}(p))$ |
| $\mathrm{Conf}^{\mathrm{max},e}(p)$ | $= \max(C_{\mathrm{mul}}\mathrm{Conf}^{\mathrm{repr},e}(p), \mathrm{Conf}^{\mathrm{reg}}(p))$ |
| $\rho_a, \rho_e > 0$ | algorithmic parameters for RSE and RLSE (Section 5) |
| $\boldsymbol{x}_i \in \mathbb{R}^d$ | feature vector for arm $i$ (RLSE, Section 7) |
| $N_i^{\mathrm{lin}}(p)$ | number of draws of arm $i$ at phase $p$ (RLSE, Section 7) |
| $p_s$ | characterizing phase for REC (equation 36) |
| $\Delta_{ij}$ | $= \mu_i - \mu_j$ |
| $\mathcal{G}^{\mathrm{reg}}$ | an event used for regret analysis (equation 55) |
| $p_{s,i}$ | characterizing phases for RSE and RLSE (equation 80, equation 81) |
| $\nu$ | model parameter such that $\mu_2 = 1/2 + \nu$ (Section G) |
| $N_C(U) \in \mathbb{N}$ | integer as a function of randomness $U$ (Section G) |
| $\mathbb{P}_{\nu,U}, \mathbb{E}_{\nu,U}$ | prob. and expect. on model $(1/2, 1/2 + \nu)$ with randomness $U$ (Section G) |
| $\mathbb{P}_\nu, \mathbb{E}_\nu$ | prob. and expect. on model $(1/2, 1/2 + \nu)$, marginalized over $U$ (Section G) |

---

**Algorithm 3:** Replicable Linear Successive Elimination (RLSE)

---

**1** Initialize the candidate set $\mathcal{A}_1 = [K]$.

**2 while** $p = 1, 2, \ldots, P$ **do**

**3**      Draw shared random variables $U_{p,i} \sim \text{Unif}(0,1)$ for each $i = 0, 1, 2, \ldots, K$.

**4**      Draw each arm $i$ for $\lceil N_p \pi_i^{*,\text{app}} \rceil$ times.        $\rightarrow$ Approximated G-optimal exploration

**5**      $\mathcal{A}_{p+1} \leftarrow \mathcal{A}_p$.

**6**      **if** $\hat{\Delta}(p) \geq (2 + U_{p,0})\text{Conf}^{\max,a}(p)$ **then**

**7**          $\mathcal{A}_{p+1} = \{\arg\max_i \hat{\mu}_i(p)\}$.        $\rightarrow$ Eliminate all arms except for one.

**8**      **for** $i \in \mathcal{A}_{p+1}$ **do**

**9**          **if** $\hat{\Delta}_i(p) \geq (2 + U_{p,i})\text{Conf}^{\max,e}(p)$ **then**

**10**             $\mathcal{A}_{p+1} \leftarrow \mathcal{A}_{p+1} \setminus \{i\}$.        $\rightarrow$ Eliminate arm $i$.

---

## A   Notation

Table 3 provides a summary of the notation we use.

### A.1   Simulation environment

The simulation took less than one hour on a Jupyterhub server on a standard desktop machine.

## B   Necessity of Randomization for a Replicable Algorithm

A trivial deterministic algorithm, such as the round-robin algorithm, is indeed replicable. Therefore, let us focus our interest on algorithms that exhibit small regret. Indeed, any no-regret algorithm is non-replicable without randomization.

**Theorem 17.** *A deterministic no-regret algorithm is non-replicable for any $\rho < 1/2$.*

*Proof of Theorem 17.* Consider the two-armed bandit problem. By assumption, the draw sequence is deterministic given the dataset $\mathcal{D} = (X_{i,t})_{i \in [K], t \in [T]}$. Since the algorithm is deterministic, there exists a 0,1-valued function $f$ over datasets such that $f(\mathcal{D}) = 1$ implies $N_1(T) \geq T/2$. Note that $\mathbb{P}[f(\mathcal{D}) = 1]$ is a continuous function of $(\mu_1, \mu_2)$, as the probability of each realization of the sequence is continuous with respect to $(\mu_1, \mu_2)$. Consider an arbitrary $\Delta > 0$. Using the fact that the algorithm has small regret, for sufficiently large $T$, we have $\mathbb{P}[f(D) = 1] > 2/3$ for the model $(1/2 + \Delta, 1/2)$ and $\mathbb{P}[f(D) = 1] < 1/3$ for the model $(1/2 - \Delta, 1/2)$. By the continuity of $\mathbb{P}[f(D) = 1]$, there exists a model $(\mu_1, 1/2)$ such that $\mathbb{P}[f(D) = 1] = 1/2$. For this model, the algorithm is **non**-$\rho$-replicable for any $\rho < 1/2$. $\qquad\square$

Note that the randomness introduced in Perchet et al. (2016) is to shuffle the allocation of arms (e.g., pull arms $1, 1, 2, 2$ or $1, 2, 1, 2$ for the exploration phase of $t = 1, 2, 3, 4$). Under the i.i.d. rewards, shuffling does not change the distribution of rewards, and thus, the results above still apply.

## C   Replicable Linear Successive Elimination

The Replicable Linear Successive Elimination (RLSE) algorithm is described in Algorithm 3.

## D   Proofs on General Bound

*Proof of Theorem 9.* Let $\mathcal{G} = \bigcap_p \mathcal{G}_p$. We have

$$\rho := \mathbb{P}_{U, \mathcal{D}^{(1)}, \mathcal{D}^{(2)}} \left[ (I_1^{(1)}, I_2^{(1)}, \ldots, I_T^{(1)}) \neq (I_1^{(2)}, I_2^{(2)}, \ldots, I_T^{(2)}) \right] \tag{23}$$

$$\leq \mathbb{P}[(I_1^{(1)}, I_2^{(1)}, \ldots, I_T^{(1)}) \neq (I_1^{(2)}, I_2^{(2)}, \ldots, I_T^{(2)}), \mathcal{G}^{(1)}, \mathcal{G}^{(2)}] + \mathbb{P}_{\mathcal{D}^{(1)}}[\mathcal{G}^c] + \mathbb{P}_{\mathcal{D}^{(2)}}[\mathcal{G}^c] \tag{24}$$

$$= \mathbb{P}[(I_1^{(1)}, I_2^{(1)}, \ldots, I_T^{(1)}) \neq (I_1^{(2)}, I_2^{(2)}, \ldots, I_T^{(2)}), \mathcal{G}^{(1)}, \mathcal{G}^{(2)}] + 2\rho^G \tag{25}$$

$$\text{(by Definition 7)} \tag{26}$$

$$\leq \mathbb{P}\left[(d_p^{(1)})_{p=1}^P \neq (d_p^{(2)})_{p=1}^P, \mathcal{G}^{(1)}, \mathcal{G}^{(2)}\right] + 2\rho^G \tag{27}$$

$$\text{(by definition of decision variables)} \tag{28}$$

$$\leq \sum_p \mathbb{P}\left[d_p^{(1)} \neq d_p^{(2)}, \cap_{p'=1}^{p-1}\left\{d_{p'}^{(1)} = d_{p'}^{(2)}\right\}, \mathcal{G}^{(1)}, \mathcal{G}^{(2)}\right] + 2\rho^G \tag{29}$$

$$\leq \sum_p \mathbb{P}\left[d_p^{(1)} \neq d_p^{(2)}, \cap_{p'=1}^{p-1}\left\{d_{p'}^{(1)} = d_{p'}^{(2)}, \mathcal{G}_{p'}^{(1)}, \mathcal{G}_{p'}^{(2)}\right\}, \mathcal{G}_p^{(1)}, \mathcal{G}_p^{(2)}\right] + 2\rho^G \tag{30}$$

$$\leq \sum_p \mathbb{P}\left[d_p^{(1)} \neq d_p^{(2)} \,\middle|\, \cap_{p'=1}^{p-1}\left\{d_{p'}^{(1)} = d_{p'}^{(2)}, \mathcal{G}_{p'}^{(1)}, \mathcal{G}_{p'}^{(2)}\right\}, \mathcal{G}_p^{(1)}, \mathcal{G}_p^{(2)}\right] + 2\rho^G \tag{31}$$

$$\leq \sum_{p,i} \mathbb{P}\left[d_{p,i}^{(1)} \neq d_{p,i}^{(2)} \,\middle|\, \cap_{p'=1}^{p-1}\left\{d_{p'}^{(1)} = d_{p'}^{(2)}, \mathcal{G}_{p'}^{(1)}, \mathcal{G}_{p'}^{(2)}\right\}, \mathcal{G}_p^{(1)}, \mathcal{G}_p^{(2)}\right] + 2\rho^G \tag{32}$$

$$\text{(by union bound)} \tag{33}$$

$$= \sum_{p,i} \rho^{(p,i)} + 2\rho^G \tag{34}$$

$$\text{(by Definition 8)}. \tag{35}$$

$$\square$$

# E   Proofs on Algorithm 1

## E.1   Stopping time

Let

$$p_{\mathrm{s}} = \min_p \left\{ p \in [P] : \mathrm{Conf}^{\max}(p) \leq \frac{10\Delta}{17} \right\}. \tag{36}$$

In the proof, we show that the algorithm is likely to break the loop at phase $p_s$ or $p_{s+1}$. Since $\mathrm{Conf}^{\max}(p) = \epsilon_p \max\left(\sqrt{\log(KTP)}, C_{\mathrm{mul}}\sqrt{\frac{\log(18K^2P/\rho)}{\rho^2}}\right)$, we can see that

$$N_{p_s} \leq 8a^2\sigma^2 \left(\frac{17}{10\Delta}\right)^2 \left(\log(KTP) + (C_{\mathrm{mul}})^2 \frac{\log(18K^2P/\rho)}{\rho^2}\right). \tag{37}$$

## E.2   Replicability of Algorithm 1

This section bounds the probability of nonreplication of Theorem 10. To do so, we use the general bound of Section 3. Let the good event (Definition 7) be

$$\mathcal{G} = \bigcap_{p \in [P]} \mathcal{G}_p \tag{38}$$

$$\mathcal{G}_p = \bigcap_{i,j \in [K]} \left\{ |\Delta_{ij} - \hat{\Delta}_{ij}(p)| \leq \rho \mathrm{Conf}^{\mathrm{repr}}(p) \right\}. \tag{39}$$

Event $\mathcal{G}$ states that all estimators lie in a region that is $\rho$ times smaller than $\mathrm{Conf}^{\mathrm{repr}}(p)$.

**Lemma 18.** *Event $\mathcal{G}$ holds with probability at least $1 - \rho/18$.*

It is easy to see that the only decision variable at phase $p$ is

$$d_{(p,0)} := \mathbf{1}\left[\hat{\Delta}(p) \geq (2 + U_p)\mathrm{Conf}^{\max}(p)\right].$$

**Lemma 19.** *Under $\mathcal{G}$, the probability of nonreplication is $\rho^{(p,0)} = 0$ for any $p \neq p_{\mathrm{s}}$.*

Lemma 19 states that the only effective decision variable is that of phase $p_{\mathrm{s}}$.

**Lemma 20.** *Under $\mathcal{G}$, the probability of nonreplication at each decision point is at most $\rho^{(p,0)} = 8\rho/9$ for $p = p_{\mathrm{s}}$.*

*Proof of nonreplicability part of Theorem 10.* Theorem 9 and Lemmas 18–20 imply that the probability of misidentification is at most $2 \times \rho/18 + 8\rho/9 = \rho$, which completes the proof. □

In the following, we derive Lemmas 18–20.

*Proof of Lemma 18.* Since $\Delta_{ij} = \mu_i - \mu_j$, $\hat{\Delta}_{ij} - \Delta_{ij}$ is estimating the sum of two $\sigma$-subgaussian random variables, which is a $2\sigma$-subgaussian random variable. By using Lemma 1 and taking a union bound over all possible $K(K-1)/2$ pairs of $ij$ and phases $1, \ldots, P$, Event $\mathcal{G}$ holds with high probability:

$$\mathbb{P}[\mathcal{G}] \geq 1 - K(K-1)P \exp\left(-\frac{(\rho \mathrm{Conf}^{\mathrm{repr}}(p))^2 N_p}{8\sigma^2}\right) \tag{40}$$

$$\geq 1 - K(K-1)P \exp\left(-\log(18K^2 P/\rho)\right) \quad \text{(by definition of } N_p, \mathrm{Conf}^{\mathrm{repr}}(p)) \tag{41}$$

$$\geq 1 - K(K-1)P \times \frac{\rho}{18K^2 P} \geq 1 - \frac{\rho}{18}, \tag{42}$$

which completes the proof. □

*Proof of Lemma 19.* First, we show that there are at most 2 phases where the break from the loop (i.e., $\min_p\{p : d_{(p,1)} = 1\}$) occurs. We first show that a break never occurs if $p < p_s$, which implies

$$\Delta < \frac{17}{10}\mathrm{Conf}^{\mathrm{max}}(p),$$

and thus

$$\hat{\Delta}(p) \leq \Delta + \rho \mathrm{Conf}^{\mathrm{repr}}(p) \quad \text{(by } \mathcal{G}) \tag{43}$$

$$\leq \Delta + \frac{\rho}{C_{\mathrm{mul}}}\mathrm{Conf}^{\mathrm{max}}(p) \tag{44}$$

$$\leq \Delta + \frac{4\rho}{9}\mathrm{Conf}^{\mathrm{max}}(p) \tag{45}$$

$$< \left(\frac{17}{10} + \frac{4\rho}{9}\right)\mathrm{Conf}^{\mathrm{max}}(p) \tag{46}$$

$$\leq 2\mathrm{Conf}^{\mathrm{max}}(p) \quad \text{(by } \rho \leq 1/2) \tag{47}$$

$$\leq (2 + U_p)\mathrm{Conf}^{\mathrm{max}}(p). \tag{48}$$

We next consider the case where $p = p_s + 1$. In this case, we have

$$\Delta \geq \frac{17}{5}\mathrm{Conf}^{\mathrm{max}}(p).$$

This implies a break, because

$$\hat{\Delta}(p) \geq \Delta - \rho \mathrm{Conf}^{\mathrm{repr}}(p) \quad \text{(by } \mathcal{G}) \tag{49}$$

$$\geq \Delta - \frac{\rho}{C_{\mathrm{mul}}}\mathrm{Conf}^{\mathrm{max}}(p) \tag{50}$$

$$\geq \frac{17}{5}\mathrm{Conf}^{\mathrm{max}}(p) - \frac{4\rho}{9}\mathrm{Conf}^{\mathrm{max}}(p) \tag{51}$$

$$\geq 3\mathrm{Conf}^{\mathrm{max}}(p) \quad \text{(by } \rho \leq 1/2) \tag{52}$$

$$\geq (2 + U_p)\mathrm{Conf}^{\mathrm{max}}(p). \tag{53}$$

The above results, combined with the fact that $\epsilon_p$ is halved at each phase, imply that the only decision points where the decision variable can take both values $\{0, 1\}$ are $p_{\mathrm{s}}, p_{\mathrm{s}} + 1$. Therefore, if the decision variable at phase $p_{\mathrm{s}}$ matches, the decision variables at $p_{\mathrm{s}} + 1$ and subsequent phases match. $\qquad\square$

*Proof of Lemma 20.* For phase $p_s$, we bound the probability of nonreplication. At the end of phase $p = p_s$, it utilizes the randomness $U_p \sim \mathrm{Unif}(0, 1)$. The random variable $(2 + U_p)\mathrm{Conf}^{\mathrm{max}}(p)$ is uniformly distributed on a region of size $\mathrm{Conf}^{\mathrm{max}}(p)$. Meanwhile, event $\mathcal{G}$ implies

$$\left| \left( \hat{\Delta}^{(1)}(p) - (2 + U_p)\mathrm{Conf}^{\mathrm{max}}(p) \right) - \left( \hat{\Delta}^{(2)}(p) - (2 + U_p)\mathrm{Conf}^{\mathrm{max}}(p) \right) \right| \leq 2\rho\mathrm{Conf}^{\mathrm{repr}}(p), \qquad (54)$$

where $\hat{\Delta}^{(1)}(p), \hat{\Delta}^{(2)}(p)$ are the corresponding quantities on the two different runs. This implication suggests that within a region of at most width $2\rho\mathrm{Conf}^{\mathrm{repr}}(p)$, the expressions $\left( \hat{\Delta}^{(1)}(p) - (2 + U_p)\mathrm{Conf}^{\mathrm{max}}(p) \right)$ and $\left( \hat{\Delta}^{(2)}(p) - (2 + U_p)\mathrm{Conf}^{\mathrm{max}}(p) \right)$ can have different signs. Therefore, the probability of nonreplication is at most

$$\frac{2\rho\mathrm{Conf}^{\mathrm{repr}}(p)}{\mathrm{Conf}^{\mathrm{max}}(p)} \leq 8\rho/9,$$

where the last inequality follows from the assumption $C_{\mathrm{mul}} \geq 9/4$. $\qquad\square$

### E.3 Regret bound of Algorithm 1

This section derives the regret bound in Theorem 10. We prepare the following events:

$$\mathcal{G}^{\mathrm{reg}} = \bigcap_{p \in [P]} \mathcal{G}^{\mathrm{reg}}_p \qquad (55)$$

$$\mathcal{G}^{\mathrm{reg}}_p = \bigcap_{i,j \in [K]} \left\{ |\Delta_{ij} - \hat{\Delta}_{ij}(p)| \leq \mathrm{Conf}^{\mathrm{reg}}(p) \right\}. \qquad (56)$$

Event $\mathcal{G}^{\mathrm{reg}}$ states that all estimators lie in the confidence region. The following theorem bounds the probability such that event $\mathcal{G}^{\mathrm{reg}}$ occurs.

**Lemma 21.**

$$\mathbb{P}[\mathcal{G}^{\mathrm{reg}}] \geq 1 - \frac{K}{T}.$$

*Proof of Lemma 21.* The proof follows similar steps as Lemma 18. By using Lemma 1 and taking a union bound over all possible $K(K-1)/2$ pairs of $ij$ and phases $1, \ldots, P$, Event $\mathcal{G}^{\mathrm{reg}}$ holds with high probability:

$$\mathbb{P}[\mathcal{G}^{\mathrm{reg}}] \geq 1 - K(K-1)P \exp\left( -\frac{(\mathrm{Conf}^{\mathrm{reg}}(p))^2 N_p}{8\sigma^2} \right) \qquad (57)$$

$$\geq 1 - K(K-1)P \exp\left( -\log(KPT) \right) \quad \text{(by definition of } N_p, \mathrm{Conf}^{\mathrm{reg}}(p)) \qquad (58)$$

$$\geq 1 - K(K-1)P \times \frac{1}{KPT} \geq 1 - \frac{K}{T}, \qquad (59)$$

which completes the proof. $\qquad\square$

Assume that $\mathcal{G}^{\mathrm{reg}}$ holds. The following shows that the break occurs by the end of phase $p_{\mathrm{s}} + 2$.

$$\hat{\Delta}(p_{\mathrm{s}} + 2) \geq \Delta - 2\mathrm{Conf}^{\mathrm{reg}}(p_{\mathrm{s}} + 2) \quad \text{(by } \mathcal{G}^{\mathrm{reg}}) \qquad (60)$$

$$\geq \Delta - 2\mathrm{Conf}^{\mathrm{max}}(p_{\mathrm{s}} + 2) \qquad (61)$$

$$\geq \frac{34}{5}\mathrm{Conf}^{\mathrm{max}}(p_{\mathrm{s}} + 2) - 2\mathrm{Conf}^{\mathrm{max}}(p_{\mathrm{s}} + 2) \quad \text{(by definition of } p_{\mathrm{s}}) \qquad (62)$$

$$\geq 3\mathrm{Conf}^{\mathrm{max}}(p) \geq (2 + U_p)\mathrm{Conf}^{\mathrm{max}}(p). \qquad (63)$$

The regret up to this phase is at most

$$\sum_i \Delta_i \times N_{p_\mathrm{s}+2}.$$

We have

$$N_{p_\mathrm{s}+2} = O\left(\log T \times a^{2(p_\mathrm{s}+2)}\right) \tag{64}$$

$$= O\left(\log T \times a^{2p_\mathrm{s}}\right) \tag{65}$$

$$= O\left(\left(\frac{1}{\Delta}\right)^2 \max\left(\log(KTP) + \frac{\log(KP/\rho)}{\rho^2}\right)\right). \quad \text{(by equation 37)} \tag{66}$$

Moreover, assume that $\hat{\mu}_i(p) \geq \hat{\mu}_1(p)$ when a break occurs at phase $p$. Then,

$$\hat{\mu}_i(p) - \hat{\mu}_1(p) \geq (2 + U_p)\mathrm{Conf}^{\max}(p) \quad \text{(by the fact that break occurs)} \tag{67}$$

$$> 2\mathrm{Conf}^{\max}(p) \tag{68}$$

$$> 2\mathrm{Conf}^{\mathrm{reg}}(p). \tag{69}$$

Meanwhile, $\mathcal{G}^{\mathrm{reg}}$ implies for all phase $p$

$$\hat{\mu}_i(p) - \hat{\mu}_1(p) \leq 2\mathrm{Conf}^{\mathrm{reg}}(p), \tag{70}$$

which implies that equation 69 never occurs. By proof of contradiction. $\hat{\mu}_1(p) > \hat{\mu}_i(p)$. Therefore, the empirically best arm is always the true best arm, and we have zero regret during the exploitation period under $\mathcal{G}^{\mathrm{reg}}$. In summary,

$$\mathbb{E}[\mathrm{Reg}(T)] \leq \mathbb{E}[\mathbf{1}[\mathcal{G}^{\mathrm{reg}}] \cdot \mathrm{Reg}(T)] + O(1) \tag{71}$$

$$\text{(equation 57 implies } \Pr[(\mathcal{G}^{\mathrm{reg}})^c] \text{ is } K/T = O(1/T)) \tag{72}$$

$$\leq \underbrace{O\left(\sum_i \Delta_i N_{p_s+2}\right)}_{\text{Regret during exploration}} + \underbrace{0}_{\substack{\text{Regret during exploitation}}} + \underbrace{O(1)}_{\substack{\text{Regret in the case of } (\mathcal{G}^{\mathrm{reg}})^c}} \tag{73}$$

$$\leq O\left(\sum_i \frac{\Delta_i}{\Delta^2} \max\left(\log(KTP) + \frac{\log(KP/\rho)}{\rho^2}\right)\right), \tag{74}$$

which, combined with $P = O(\log T)$, completes the proof.

## F    Proofs on Algorithm 2

### F.1    Replicability of Algorithm 2

Similarly to that of Theorem 10, we define the good event. Let the good event (Definition 7) be

$$\mathcal{G} = \bigcap_{p \in [P]} \left(\mathcal{G}_{p,0} \cap \bigcap_{i \in [K]} \mathcal{G}_{p,i}\right) \tag{75}$$

$$\mathcal{G}_{a,0} = \bigcap_{i,j \in [K]} \left\{|\Delta_{ij} - \hat{\Delta}_{ij}(p)| \leq \rho_a \mathrm{Conf}^{\mathrm{repr},a}(p)\right\} \tag{76}$$

$$\mathcal{G}_{a,i} = \bigcap_{i,j \in [K]} \left\{|\Delta_{ij} - \hat{\Delta}_{ij}(p)| \leq \rho_e \mathrm{Conf}^{\mathrm{repr},e}(p)\right\}. \tag{77}$$

**Lemma 22.** *Event $\mathcal{G}$ holds with probability at least $1 - \rho/18$. Moreover, under $\mathcal{G}$, arm 1 (= best arm) is never eliminated (i.e., $1 \in \mathcal{A}_p$ for all $p$).*

There are $K + 1$ binary decision variables at phase $p$. The first decision variable is

$$d_{(p,0)} := \mathbf{1}\left[\hat{\Delta}(p) > (2 + U_{p,0})\text{Conf}^{\max,a}(p)\right]. \tag{78}$$

The other $K$ decision variables are

$$d_{(p,i)} := \mathbf{1}\left[\hat{\Delta}_i(p) > (2 + U_{p,i})\text{Conf}^{\max,e}(p)\right] \tag{79}$$

for each $i = 1, 2, \ldots, K$. If all decision variables are identical between two runs then the sequence of draws is identical between them.

**Lemma 23.** *Let*

$$p_{\text{s},0} = \min_p \left\{ p \in [P] : \text{Conf}^{\max,a}(p) \leq \frac{10\Delta}{17} \right\}. \tag{80}$$

*Under $\mathcal{G}$, we have $\rho^{(p,0)} = 0$ for any $p \neq p_{\text{s},0}$.*

**Lemma 24.** *Let*

$$p_{\text{s},i} = \min_p \left\{ p \in [P] : \text{Conf}^{\max,e}(p) \leq \frac{10\Delta_i}{17} \right\}, \tag{81}$$

*for each $i \in [K]$. Then, $\rho^{(p,i)} = 0$ for all $p \neq p_{\text{s},i}$.*

**Lemma 25.** *Under $\mathcal{G}$, the probability of nonreplication at each decision point is at most $\rho^{(p,0)} = 8\rho_a/9$ or $\rho^{(p,i)} = 8\rho_e/9$ for $i \in \{2, \ldots, K\}$.*

*Proof of nonreplicability part of Theorem 11.* Theorem 9 and Lemmas 22–25 imply that the probability of misidentification is at most $(\rho_a + (K-1)\rho_e) \times (2 \times \frac{1}{18} + \frac{8}{9}) = \rho$, which completes the proof. □

*Proof of Lemma 22.* By using essentially the same discussion as Lemma 18 yields the fact that event $\mathcal{G}$ holds with probability at least $1 - \rho/18$.

In the following, we show that arm 1 is never eliminated under $\mathcal{G}$. Elimination of arm 1 at phase $p$ due to $\rho_a$ implies that there exists a suboptimal arm $i \neq 1$ such that

$$\hat{\mu}_i(p) - \mu_1(p) \geq (2 + U_{p,0})\text{Conf}^{\max,a}(p) \geq 2\text{Conf}^{\max,a}(p), \tag{82}$$

which never occurs under $\mathcal{G}$, since $\mathcal{G}$ implies

$$|\mu_1 - \hat{\mu}_1(p)|, |\mu_i - \hat{\mu}_i(p)| < \text{Conf}^{\text{repr},a}(p) \leq \text{Conf}^{\max,a}(p)$$

and thus

$$\hat{\mu}_i(p) - \mu_1(p) < 2\text{Conf}^{\max,a}(p) + \mu_i - \mu_1 \leq 2\text{Conf}^{\max,a}(p), \tag{83}$$

which contradicts equation 82. The same discussion goes for the elimination due to $\rho_e$. □

*Proof of Lemma 23.* The proof proceeds similarly to that of Lemma 19, but for the decision variable $d_{(p,0)}$. We first show that for any $p < p_{\text{s},0}$, a break (i.e., $d_{(p,0)} = 1$) never occurs. By the definition of $p_{\text{s},0}$, for $p < p_{\text{s},0}$, we have $\text{Conf}^{\max,a}(p) > \frac{10\Delta}{17}$. Under $\mathcal{G}$, we have

$$\hat{\Delta}(p) \leq \Delta + \rho_a \text{Conf}^{\text{repr},a}(p) \leq \Delta + \frac{\rho_a}{C_{\text{mul}}}\text{Conf}^{\max,a}(p) \leq \Delta + \frac{4\rho_a}{9}\text{Conf}^{\max,a}(p).$$

Since $\Delta < \frac{17}{10}\text{Conf}^{\max,a}(p)$, we get

$$\hat{\Delta}(p) < \left(\frac{17}{10} + \frac{4\rho_a}{9}\right)\text{Conf}^{\max,a}(p) \leq 2\text{Conf}^{\max,a}(p) \leq (2 + U_{p,0})\text{Conf}^{\max,a}(p),$$

where the last inequality uses $\rho_a \leq 1/2$ and $U_{p,0} \sim \text{Unif}(0,1)$. Thus, $d_{(p,0)} = 0$ for $p < p_{\text{s},0}$.

We next consider the case where $p = p_{\mathrm{s},0} + 1$. In this case, we have

$$\Delta \geq \frac{17}{5}\mathrm{Conf}^{\mathrm{max},a}(p).$$

This implies a break, because

$$\hat{\Delta}(p) \geq \Delta - \rho\mathrm{Conf}^{\mathrm{repr},a}(p) \quad (\text{by } \mathcal{G}) \tag{84}$$

$$\geq \Delta - \frac{\rho}{C_{\mathrm{mul}}}\mathrm{Conf}^{\mathrm{max},a}(p) \tag{85}$$

$$\geq \frac{17}{5}\mathrm{Conf}^{\mathrm{max},a}(p) - \frac{4\rho}{9}\mathrm{Conf}^{\mathrm{max},a}(p) \tag{86}$$

$$\geq 3\mathrm{Conf}^{\mathrm{max},a}(p) \quad (\text{by } \rho \leq 1/2) \tag{87}$$

$$\geq (2 + U_{p,0})\mathrm{Conf}^{\mathrm{max},a}(p). \tag{88}$$

In summary, the break occurs at $p = p_{\mathrm{s},0}$ or $p = p_{\mathrm{s},0} + 1$. Therefore, if the decision variable at phase $p_{\mathrm{s},0}$ matches, the decision variables at $p_{\mathrm{s},0} + 1$ and subsequent phases match. Thus, $\rho^{(p,0)} = 0$ for all $p \neq p_{\mathrm{s},0}$, as claimed. $\qquad\square$

*Proof of Lemma 24.* We proceed similarly to the proof of Lemma 23, but for the decision variable $d_{(p,i)}$ for each $i \in [K]$.

For $p < p_{\mathrm{s},i}$, by the definition of $p_{\mathrm{s},i}$, $\mathrm{Conf}^{\mathrm{max},e}(p) > \frac{10\Delta_i}{17}$. Under $\mathcal{G}$, we have

$$\hat{\Delta}_i(p) \leq \Delta_i + \rho_e\mathrm{Conf}^{\mathrm{repr},e}(p) \leq \Delta_i + \frac{\rho_e}{C_{\mathrm{mul}}}\mathrm{Conf}^{\mathrm{max},e}(p) \leq \Delta_i + \frac{4\rho_e}{9}\mathrm{Conf}^{\mathrm{max},e}(p).$$

Since $\Delta_i < \frac{17}{10}\mathrm{Conf}^{\mathrm{max},e}(p)$, we get

$$\hat{\Delta}_i(p) < \left(\frac{17}{10} + \frac{4\rho_e}{9}\right)\mathrm{Conf}^{\mathrm{max},e}(p) \leq 2\mathrm{Conf}^{\mathrm{max},e}(p) \leq (2 + U_{p,i})\mathrm{Conf}^{\mathrm{max},e}(p),$$

where the last inequality uses $\rho_e \leq 1/2$ and $U_{p,i} \sim \mathrm{Unif}(0,1)$. Thus, $d_{(p,i)} = 0$ for $p < p_{\mathrm{s},i}$.

We next consider the case where $p = p_{\mathrm{s},i} + 1$. In this case, we have

$$\Delta_i \geq \frac{17}{5}\mathrm{Conf}^{\mathrm{max},e}(p).$$

This implies a break, because

$$\hat{\Delta}_i(p) \geq \Delta_i - \rho_e\mathrm{Conf}^{\mathrm{repr},e}(p) \quad (\text{by } \mathcal{G}) \tag{89}$$

$$\geq \Delta_i - \frac{\rho_e}{C_{\mathrm{mul}}}\mathrm{Conf}^{\mathrm{max},e}(p) \tag{90}$$

$$\geq \frac{17}{5}\mathrm{Conf}^{\mathrm{max},e}(p) - \frac{4\rho_e}{9}\mathrm{Conf}^{\mathrm{max},e}(p) \tag{91}$$

$$\geq 3\mathrm{Conf}^{\mathrm{max},e}(p) \quad (\text{by } \rho_e \leq 1/2) \tag{92}$$

$$\geq (2 + U_{p,i})\mathrm{Conf}^{\mathrm{max},e}(p). \tag{93}$$

In summary, the break occurs at $p = p_{\mathrm{s},i}$ or $p = p_{\mathrm{s},i} + 1$. Therefore, if the decision variable at phase $p_{\mathrm{s},i}$ matches, the decision variables at $p_{\mathrm{s},i} + 1$ and subsequent phases match. Thus, $\rho^{(p,i)} = 0$ for all $p \neq p_{\mathrm{s},i}$, as claimed. $\qquad\square$

*Proof of Lemma 25.* We bound the probability that the decision variable $d_{(p,0)}$ (or $d_{(p,i)}$) differs between two runs under $\mathcal{G}$. The proof is similar to that of Lemma 20, but we need to consider the decision variables $d_{(p,0)}$ as well as $d_{(p,i)}$ for each $i \in [K]$.

We first consider the decision variable $d_{(p,0)}$. For phase $p_{s,0}$, we bound the probability of nonreplication. At the end of phase $p = p_{s,0}$, it utilizes the randomness $U_{p,0} \sim \text{Unif}(0,1)$. The random variable $(2 + U_{p,0})\text{Conf}^{\max,a}(p)$ is uniformly distributed on a region of size $\text{Conf}^{\max,a}(p)$. Meanwhile, event $\mathcal{G}$ implies

$$\left|\left(\hat{\Delta}^{(1)}(p) - (2 + U_{p,0})\text{Conf}^{\max,a}(p)\right) - \left(\hat{\Delta}^{(2)}(p) - (2 + U_{p,0})\text{Conf}^{\max,a}(p)\right)\right| \leq 2\rho_a \text{Conf}^{\text{repr},a}(p), \quad (94)$$

where $\hat{\Delta}^{(1)}(p), \hat{\Delta}^{(2)}(p)$ are the corresponding quantities on the two different runs. This implication suggests that within a region of at most width $2\rho_a \text{Conf}^{\text{repr},a}(p)$, the expressions $\left(\hat{\Delta}^{(1)}(p) - (2 + U_{p,0})\text{Conf}^{\max,a}(p)\right)$ and $\left(\hat{\Delta}^{(2)}(p) - (2 + U_{p,0})\text{Conf}^{\max,a}(p)\right)$ can have different signs. Therefore, the probability of nonreplication is at most

$$\frac{2\rho_a \text{Conf}^{\text{repr},a}(p)}{\text{Conf}^{\max,a}(p)} \leq 8\rho_a/9,$$

where the last inequality follows from the assumption $C_{\text{mul}} \geq 9/4$.

We next consider the decision variable $d_{(p,i)}$ for $i = 1, \ldots, K$. For phase $p_{s,i}$, we bound the probability of nonreplication. At the end of phase $p = p_{s,i}$, it utilizes the randomness $U_{p,i} \sim \text{Unif}(0,1)$. The random variable $(2 + U_{p,i})\text{Conf}^{\max,e}(p)$ is uniformly distributed on a region of size $\text{Conf}^{\max,e}(p)$. Meanwhile, event $\mathcal{G}$ implies

$$\left|\left(\hat{\Delta}_i^{(1)}(p) - (2 + U_{p,i})\text{Conf}^{\max,e}(p)\right) - \left(\hat{\Delta}_i^{(2)}(p) - (2 + U_{p,i})\text{Conf}^{\max,e}(p)\right)\right| \leq 2\rho_e \text{Conf}^{\text{repr},e}(p), \quad (95)$$

where $\hat{\Delta}_i^{(1)}(p), \hat{\Delta}_i^{(2)}(p)$ are the corresponding quantities on the two different runs. This implication suggests that within a region of at most width $2\rho_e \text{Conf}^{\text{repr},e}(p)$, the expressions $\left(\hat{\Delta}_i^{(1)}(p) - (2 + U_{p,i})\text{Conf}^{\max,e}(p)\right)$ and $\left(\hat{\Delta}_i^{(2)}(p) - (2 + U_{p,i})\text{Conf}^{\max,e}(p)\right)$ can have different signs. Therefore, the probability of nonreplication is at most

$$\frac{2\rho_e \text{Conf}^{\text{repr},e}(p)}{\text{Conf}^{\max,e}(p)} \leq 8\rho_e/9,$$

where the last inequality follows from the assumption $C_{\text{mul}} \geq 9/4$. $\qquad\square$

### F.2 Regret bound of Algorithm 2

We prepare the following events:

$$\mathcal{G}^{\text{reg}} = \bigcap_{p \in [P]} \mathcal{G}_p^{\text{reg}} \quad (96)$$

$$\mathcal{G}_p^{\text{reg}} = \bigcap_{i,j \in [K]} \left\{|\Delta_{ij} - \hat{\Delta}_{ij}(p)| \leq \text{Conf}^{\text{reg}}(p)\right\}. \quad (97)$$

Event $\mathcal{G}^{\text{reg}}$ states that all estimators lie in the confidence region.

**(A) The $O(K)$ regret bound:** This part is very identical to that of Algorithm 1 because, under $\mathcal{G}^{\text{reg}}$, all but arm 1 is eliminated by phase $p_{s,0} + 2$. We omit the proof to avoid repetition.

**(B) The other two regret bounds:** We first derive the distribution-dependent bound. We show that under $\mathcal{G}^{\text{reg}}$, each arm $i$ is eliminated by phase $p_{s,i} + 2$. Assume that $\mathcal{G}^{\text{reg}}$ holds. The following shows that the break occurs by the end of phase $p_{s,i} + 2$.

$$\hat{\mu}_i(p_{s,i} + 2) - \hat{\mu}_1(p_{s,i} + 2) \geq \Delta_i - 2\text{Conf}^{\text{reg}}(p_{s,i} + 2) \quad (\text{by } \mathcal{G}^{\text{reg}}) \quad (98)$$

$$\geq \Delta_i - 2\text{Conf}^{\max,e}(p_{s,i} + 2) \quad (99)$$

$$\geq \frac{34}{5}\text{Conf}^{\max,e}(p_{s,i} + 2) - 2\text{Conf}^{\max,e}(p_{s,i} + 2) \quad (\text{by definition of } p_{s,i}) \quad (100)$$

$$\geq 3\text{Conf}^{\max,e}(p) \geq (2 + U_p)\text{Conf}^{\max}(p). \quad (101)$$

Therefore, each arm $i$ is drawn at most

$$N_{i,\max} := O\left(4^{p_{s,i}+2}\log T\right) \tag{102}$$

$$= O\left(\frac{1}{\Delta_i^2}\left(\log(KTP) + \frac{\log(KP/\rho_e)}{(\rho_e)^2}\right)\right) \tag{103}$$

$$= O\left(\frac{1}{\Delta_i^2}\left(\log(KTP) + \frac{K^2\log(KP/\rho)}{\rho^2}\right)\right) \tag{104}$$

times.

Moreover, assume that $\hat{\mu}_i(p) \geq \hat{\mu}_1(p)$ when a break occurs at phase $p$. Let $U_p = \min_{i \in \{0\} \cup [K]} U_{p,i}$. Then,

$$\hat{\mu}_i(p) - \hat{\mu}_1(p) > 2\mathrm{Conf}^{\mathrm{reg}}(p). \tag{105}$$

Meanwhile, $\mathcal{G}^{\mathrm{reg}}$ implies for all phase $p$

$$\hat{\mu}_i(p) - \hat{\mu}_1(p) \leq 2\mathrm{Conf}^{\mathrm{reg}}(p), \tag{106}$$

which implies that equation 105 never occurs.

The regret is bounded as

$$\mathbb{E}[\mathrm{Reg}(T)] \leq \mathbb{E}[\mathbf{1}[\mathcal{G}^{\mathrm{reg}}] \cdot \mathrm{Reg}(T)] + O(1) \tag{107}$$

$$\leq \underbrace{\sum_i \Delta_i N_{i,\max}}_{\text{Regret during exploration}} + \underbrace{0}_{\text{Regret during exploitation}} + \underbrace{O(1)}_{\text{Regret in the case of }(\mathcal{G}^{\mathrm{reg}})^c} \tag{108}$$

$$\leq O\left(\sum_i \frac{1}{\Delta_i}\left(\log(KTP) + \frac{K^2\log(KP/\rho)}{\rho^2}\right)\right) \tag{109}$$

$$\leq O\left(\sum_i \frac{1}{\Delta_i}\left((\log T) + \frac{K^2\log(K(\log T)/\rho)}{\rho^2}\right)\right), \tag{110}$$

which is the second regret bound of Theorem 11.

We finally derive the distribution-independent regret bound. Letting $N_i(T)$ be the number of draws of arm $i$ in the $T$ rounds, we have

$$\mathrm{Reg}(T)\mathbf{1}[\mathcal{G}^{\mathrm{reg}}] \leq \sum_i \Delta_i N_i(T) + O(1) \tag{111}$$

$$= \sum_i \Delta_i \sqrt{N_{i,\max}}\sqrt{N_i(T)} + O(1), \tag{112}$$

and

$$\sum_i \Delta_i \sqrt{N_{i,\max}}\sqrt{N_i(T)} \leq O(1) \times \sum_i \Delta_i \sqrt{\frac{1}{\Delta_i^2}\left((\log T) + \frac{K^2\log(K(\log T)/\rho)}{\rho}\right)}\sqrt{N_i(T)} \tag{113}$$

$$\text{(by equation 104)} \tag{114}$$

$$\leq O(1) \times \sum_i \sqrt{(\log T) + \frac{K^2\log(K(\log T)/\rho)}{\rho}}\sqrt{N_i(T)} \tag{115}$$

$$\leq O(1) \times \sqrt{(\log T) + \frac{K^2\log(K(\log T)/\rho)}{\rho}}\sqrt{KT} \tag{116}$$

$$\text{(by Cauchy-Schwarz and } \sum_i N_i(T) = T), \tag{117}$$

and thus

$$\mathbb{E}[\text{Reg}(T)] \leq \mathbb{E}[\text{Reg}(T)\mathbf{1}[\mathcal{G}^{\text{reg}}]] + O(1) \tag{118}$$

$$= O\left(\sqrt{KT\left((\log T) + \frac{K^2 \log(K(\log T)/\rho)}{\rho}\right)}\right), \tag{119}$$

which is the third regret bound of Theorem 11.

## G   Proofs on the Lower Bound

In the following, we derive Theorem 12. The proof is inspired by Theorem 7.2 of Impagliazzo et al. (2022) but is significantly more challenging due to the adaptiveness of sampling. In particular, Lemma 26 utilizes the change-of-measure argument and works even if the number of draws $N_2(T)$ on arm 2 is a random variable.

*Proof of Theorem 12.* The goal of the proof here is to derive the inequality:

$$\mathbb{E}[\text{Reg}(T)] = \Omega\left(\min\left(\frac{1}{\rho^2 \Delta \log((\rho\Delta)^{-1})}, \Delta T\right)\right). \tag{120}$$

We consider the set of models $\mathcal{P}$, where $\mu_1 = 1/2$ is fixed and $\mu_2 \in [1/2 - \Delta, 1/2 + \Delta]$. With a slight abuse of notation, we specify a model in $\mathcal{P}$ by $\mu_2 - 1/2$. We also denote $U$ to the internal randomness. For example,

$$\mathbb{P}_{-\Delta,U}[\mathcal{X}]$$

be the probability that event $\mathcal{X}$ occurs under the corresponding model $(\mu_1, \mu_2) = (1/2, 1/2 - \Delta)$ and randomness $U$. Moreover, let

$$\mathbb{P}_{-\Delta}[\mathcal{X}] = \int \mathbb{P}_{-\Delta,U}[\mathcal{X}]dP(U)$$

be the probability marginalized over randomness $U$.

Let

$$p_c = \min\left(\mathbb{P}_{-\Delta}\left[\mathbb{E}_{-\Delta,U}[\text{Reg}(T)] \leq \frac{T\Delta}{8}\right], \mathbb{P}_{\Delta}\left[\mathbb{E}_{\Delta,U}[\text{Reg}(T)] \leq \frac{T\Delta}{8}\right]\right). \tag{121}$$

Namely, $p_c$ represents the ratio of randomness where the expected regret is less than $T\Delta/8$ for both the $-\Delta$ and $\Delta$ models.

If $p_c \leq 3/4$, then at least one of the models $(1/2, 1/2 - \Delta)$ or $(1/2, 1/2 + \Delta)$, for $1/4$ of the randomness $U$, the regret is larger than $\frac{T\Delta}{8}$. Therefore, the regret averaged over the randomness is larger than $\frac{T\Delta}{32}$ for that model, which completes the proof.

For the rest of the proof, we exclusively consider the case of $p_c > 3/4$. This implies that the probability of getting $U$ such that

$$\mathbb{E}_{-\Delta,U}[\text{Reg}(T)], \mathbb{E}_{\Delta,U}[\text{Reg}(T)] \leq \frac{T\Delta}{8} \tag{122}$$

is at least half ($= 1 - 2 \times (1 - 3/4)$).   In parts (A) and (B), we fix $U$ such that equation 122 holds. Part (C) marginalizes it over $U$.

**(A) Fix the randomness $U$ and consider behavior of algorithm for different models:**

Let $N_C(U)$ be the smallest among the integer $n$ such that

$$\mathbb{P}_{-\Delta,U}\left[N_2(T) \geq n\right] \leq \frac{1}{4} \text{ and} \tag{123}$$

$$\mathbb{P}_{\Delta,U}\left[N_2(T) \geq n\right] \geq \frac{3}{4}. \tag{124}$$

At least one such an integer exists; $n = T/2$ satisfies this condition because otherwise

$$\max\{\mathbb{E}_{-\Delta,U}[\text{Reg}(T)], \mathbb{E}_{\Delta,U}[\text{Reg}(T)]\} \tag{125}$$

$$\geq \max\left\{\mathbb{P}_{-\Delta,U}\left[N_2(T) \geq \frac{T}{2}\right]\frac{T\Delta}{2}, \left(1 - \mathbb{P}_{\Delta,U}\left[N_2(T) \geq \frac{T}{2}\right]\right)\frac{T\Delta}{2}\right\} \tag{126}$$

$$> \frac{T\Delta}{8} \tag{127}$$

$$\left(\text{by } \mathbb{P}_{-\Delta,U}\left[N_2(T) \geq \frac{T}{2}\right] > \frac{1}{4} \text{ or } \mathbb{P}_{\Delta,U}\left[N_2(T) \geq \frac{T}{2}\right] < \frac{3}{4}\right), \tag{128}$$

which violates equation 122.

By continuity, there exists $\nu = \nu(U) \in (-\Delta, +\Delta)$ such that $\mathbb{P}_{\nu,U}[N_2(T) \geq N_C(U)] = 1/2$. By Lemma 26, there exists $C_I = \Theta(1)$ such that, for any $\nu' \in [\nu - C_I/\sqrt{\log(N_C(U))N_C(U)}, \nu + C_I/\sqrt{\log(N_C(U))N_C(U)}] =: \mathcal{I}(U)$ we have $\mathbb{P}_{\nu',U}[N_2(T) \geq N_C(U)] \in (1/3, 2/3)$.

**(B) Marginalize it over models:** Assume that we first draw a model uniformly random from $[-\Delta, \Delta]$, and then run the algorithm. Conditioned on the shared randomness $U$, with probability at least $C_I/(\sqrt{\log(N_C(U))N_C(U)}\Delta)$, we draw model in $\mathcal{I}(U)$. For a model in $\mathcal{I}(U)$, there is $2 \times (1/3) \times (2/3) = 4/9$ probability of nonreplicability.

**(C) Marginalize it over shared random variable $U$:** The fact that $p_c > 3/4$ implies that the probability of getting $U$ such that discussions (A) and (B) hold is at least $1/2$. We let the set of such randomness as $\mathcal{U}_{half}$. Let $n_c$ be the expected value of $N_C(U)$ marginalized over[10] the distribution on $\mathcal{U}_{half}$. We have

$$\frac{1}{2} \times \frac{4}{9}\frac{C_I}{\sqrt{\log(n_c)n_c}\Delta} \tag{129}$$

$$\leq \int_{U \in \mathcal{U}_{half}} \frac{4}{9}\frac{C_I}{\sqrt{\log(n_c)n_c}\Delta}dP(U) \tag{130}$$

$$\text{(by at least half of } U \text{ are in } \mathcal{U}_{half}) \tag{131}$$

$$\leq \int_{U \in \mathcal{U}_{half}} \frac{4}{9}\frac{C_I}{\sqrt{\log(N_C(U))N_C(U)}\Delta}dP(U), \tag{132}$$

where the last transformation applied Jensen's inequality $f(\mathbb{E}[x]) \leq \mathbb{E}[f(x)]$. Here, we used a convex function $f(x) = 1/\sqrt{x\log x}$, $x = N_C(U)$ and the expectation $n_c = \mathbb{E}[x]$ was taken on the randomness of $U$ over $\mathcal{U}_{half}$.

Moreover, by part (B) and $\rho$-replicability, we obtain

$$\int_{U \in \mathcal{U}_{half}} \frac{4}{9}\frac{C_I}{\sqrt{\log(N_C(U))N_C(U)}\Delta}dP(U) \leq \rho, \tag{133}$$

which implies

$$n_c = \Omega\left(\frac{1}{(\rho\Delta)^2\log((\rho\Delta)^{-1})}\right). \tag{134}$$

The regret is lower-bounded as

$$\max\{\mathbb{E}_{-\Delta}[\text{Reg}(T)], \mathbb{E}_{\Delta}[\text{Reg}(T)]\} \tag{135}$$

$$\geq \max\left\{\mathbb{P}_{-\Delta}[N_2(T) \geq n_c - 1]\Delta(n_c - 1), (1 - \mathbb{P}_{-\Delta}[N_2(T) \geq n_c - 1])\Delta(T - n_c + 1)\right\} \tag{136}$$

$$\geq \max\left\{\mathbb{P}_{-\Delta}[N_2(T) \geq n_c - 1]\Delta(n_c - 1), (1 - \mathbb{P}_{-\Delta}[N_2(T) \geq n_c - 1])\Delta\left(\frac{T}{2} + 1\right)\right\} \tag{137}$$

---

[10]The discussion in the next display, which utilizes the convexity of the function, is also useful in fixing a minor error of Lemma 7.2 in Impagliazzo et al. (2022), which implicitly assumes that the sample complexity ($m$ therein) is independent of the randomness ($r$ therein).

$$\text{(by } n_c \leq T/2)\tag{138}$$

$$\geq \frac{1}{4} \times \Delta(n_c - 1)\tag{139}$$

$$\text{(by } N_C(U) - 1 \text{ violates equation 123 or equation 124, and } n_c \leq T/2)\tag{140}$$

$$= \Omega\left(\frac{1}{\rho^2 \Delta \log((\rho\Delta)^{-1})}\right). \quad \text{(by equation 134)}\tag{141}$$

$\square$

**Remark 2.** (Extension for $K$-armed Lower Bound) *Here, equation 135 bounds at least one of two quantities* $\mathbb{E}_{-\Delta}[\mathrm{Reg}(T)]$ *or* $\mathbb{E}_{\Delta}[\mathrm{Reg}(T)]$. *If we can directly bound* $\mathbb{E}_{-\Delta}[\mathrm{Reg}(T)]$, *then we should be able to extend the results here to the* $K$-armed case.

### G.1 Lemmas for regret lower bound

The following lemma is used to bound the gradient of the probability of occurrences. This lemma corresponds to the derivative of the acceptance function[11] in Lemma 7.2 of Impagliazzo et al. (2022), but more technical due to the fact that $N_2(T)$ is a random variable.

**Lemma 26.** (Likelihood ratio) *Let*

$$\mathcal{E} = \{N_2(T) \leq N_C\}\tag{142}$$

*and* $\nu$ *be such that* $\mathbb{P}_{\nu,U}[\mathcal{E}] = 1/2$. *There exists a value* $C_I = \Theta(1)$ *that does not depend on the shared random variable* $U$ *such that, for any model* $\nu' \in [\nu - C_I/\sqrt{\log(N_C)N_C}, \nu + C_I/\sqrt{\log(N_C)N_C}]$, *we have*

$$\mathbb{P}_{\nu',U}[\mathcal{E}] \in \left(\frac{1}{3}, \frac{2}{3}\right).\tag{143}$$

*Proof of Lemma 26.* In the following, we bound $\mathbb{P}_{\nu',U}[\mathcal{E}]$ by using the change-of-measure argument.

Let the log-likelihood ratio between the models $\nu, \nu'$ be[12]

$$L_t = \sum_{n=1}^{N_2(t)} \log\left(\frac{r_{2,n}\left(\frac{1}{2} + \nu\right) + (1 - r_{2,n})\left(\frac{1}{2} - \nu\right)}{r_{2,n}\left(\frac{1}{2} + \nu'\right) + (1 - r_{2,n})\left(\frac{1}{2} - \nu'\right)}\right),\tag{144}$$

where $r_{2,n}$ is the $n$-th reward from arm 2.

We have

$$\mathbb{P}_{\nu',U}[\mathcal{E}] = \mathbb{E}_{\nu,U}\left[\mathbf{1}[\mathcal{E}]e^{-L_T}\right]. \quad \text{(change-of-measure)}\tag{145}$$

Note that, under $\nu$ the random variable

$$\log\left(\frac{r_{2,n}\left(\frac{1}{2} + \nu\right) + (1 - r_{2,n})\left(\frac{1}{2} - \nu\right)}{r_{2,n}\left(\frac{1}{2} + \nu'\right) + (1 - r_{2,n})\left(\frac{1}{2} - \nu'\right)}\right)$$

is mean $d_{\mathrm{KL}}(\nu, \nu')$ and bounded by

$$R = \max\left(\left|\log\left(\frac{2 + \nu}{2 + \nu'}\right)\right|, \left|\log\left(\frac{2 - \nu}{2 - \nu'}\right)\right|\right) = O(|\nu - \nu'|) = O\left(\frac{C_I}{\sqrt{\log(N_C)N_C}}\right).$$

Here, $d_{\mathrm{KL}}(p, q)$ is the KL divergence between two Bernoulli distributions with parameters $p, q \in (0, 1)$. Under $\mathcal{E}$, $N_2(T) \leq N_C$ and $L_T$ is bounded as the max of random variables

$$L_T \leq \max_{N \leq N_C}\left(\sum_{n \leq N} Z_n\right),$$

---

[11]Namely, $\mathrm{ACC}(p)$ therein.
[12]On these models, $\mu_1 = 1/2$, $\mu_2 = 1/2 + \nu$ or $1/2 + \nu'$.

where

$$Z_n := \log \left( \frac{r_{2,n} \left( \frac{1}{2} + \nu \right) + (1 - r_{2,n}) \left( \frac{1}{2} - \nu \right)}{r_{2,n} \left( \frac{1}{2} + \nu' \right) + (1 - r_{2,n}) \left( \frac{1}{2} - \nu' \right)} \right)$$

is a random variable with its mean $d_{\mathrm{KL}}(\nu, \nu')$ and radius $R$. Hoeffding inequality and union bound over $N = 1, 2, \ldots, N_C$ implies that, with probability at least $1 - 1/12$ we have

$$|L_T| \le N_C d_{\mathrm{KL}}(1/2 + \nu, 1/2 + \nu') + R\sqrt{\log(2 \times 12 \times N_C)N_C/2} \tag{146}$$

$$= O \left( N_C \times \frac{C_I^2}{\log(N_C)N_C} \right) + O \left( \frac{C_I}{\sqrt{\log(N_C)N_C}} \times \sqrt{N_C \log(N_C)} \right) = O(C_I). \tag{147}$$

Setting an appropriate width $C_I = \Theta(1)$ guarantees that $e^{-L_T} \in [1 - 1/6, 1 + 1/6]$ with probability at least $1 - 1/12$, which, together with the change-of-measure implies equation 143. $\qquad\square$

## H  Proofs on Algorithm 3

### H.1  Replicability of Algorithm 3

We omit the derivation of the nonreplicability bound of Algorithm 3 because it is very similar to that of Algorithm 2. The only difference is that the amount of exploration is based on a G-optimal design, but its confidence bound of equation 20 suffices to derive the good event that is identical to equation 38.

### H.2  Regret bound of Algorithm 3

This section shows the regret bounds of Theorem 16. The main difference from Algorithm 2 is that the number of samples for each arm is $N_i^{\mathrm{lin}}(p)$ that satisfies $\sum_i N_i^{\mathrm{lin}}(p) \le N^{\mathrm{lin}}(p) + K$.

**(A) The $O(d)$ regret bound:** We derive the first regret bound in Theorem 16. We first derive the distribution-dependent bound. Similar discussion to Lemma 25 states that, under $\mathcal{G}^{\mathrm{reg}}$, Line 6 in Algorithm 3 eliminates all but the best arm by phase $p_{\mathrm{s},0} + 2$, where $p_{\mathrm{s},0}$ is identical to that of Algorithm 2. The regret is bounded as

$$\mathbb{E}[\mathrm{Reg}(T)] \tag{148}$$

$$\le \mathbb{E}[\mathbf{1}[\mathcal{G}^{\mathrm{reg}}]\mathrm{Reg}(T)] + O(1) \tag{149}$$

$$\le O \underbrace{\left( \sum_i N_i^{\mathrm{lin}}(p_{\mathrm{s},0} + 2) \right)}_{\text{Regret during exploration}} + \underbrace{0}_{\text{Regret during exploitation}} + \underbrace{O(1)}_{\text{Regret in the case of } (\mathcal{G}^{\mathrm{reg}})^c} \tag{150}$$

$$= O \left( \frac{\sigma^2 d}{(\epsilon_{p_{\mathrm{s},0}})^2} + KP \right) \tag{151}$$

$$= O \left( \frac{\sigma^2 d}{\Delta^2} \left( \log(KTP) + \frac{\log(KP/\rho)}{\rho^2} \right) + KP \right) \tag{152}$$

$$= O \left( \frac{\sigma^2 d}{\Delta^2} \left( \log T + \frac{\log(K(\log T)/\rho)}{\rho^2} \right) \right), \tag{153}$$

which is the first regret bound of Theorem 16.

**(B) The distribution-independent regret bound:**

Similar discussion as Algorithm 2 states that, under $\mathcal{G}$, arm $i$ is eliminated by $p_{\mathrm{s},i} + 2$. The regret is bounded as

$$\mathrm{Reg}(T)\mathbf{1}[\mathcal{G}] \le \sum_i \Delta_i N_i(T) + O(1) \tag{154}$$

$$\leq \sum_i \Delta_i \sqrt{\sum_{p \leq p_{\mathrm{s},i}+2} N_i^{\mathrm{lin}}(p)} \sqrt{N_i(T)} + O(1), \tag{155}$$

and

$$\sum_i \Delta_i \sqrt{\sum_{p \leq p_{\mathrm{s},i}+2} N_i^{\mathrm{lin}}(p)} \sqrt{N_i(T)} \tag{156}$$

$$\leq O(1) \times \sum_i \Delta_i \sqrt{\frac{d}{\Delta_i^2}\left(\log T + \frac{K^2 \log(K(\log T)/\rho)}{\rho^2}\right)} \sqrt{N_i(T)} \tag{157}$$

$$\text{(by equation 19)} \tag{158}$$

$$\leq O(1) \times \sum_i \sqrt{d\left(\log T + \frac{K^2 \log(K(\log T)/\rho)}{\rho^2}\right)} \sqrt{N_i(T)} \tag{159}$$

$$\leq O(1) \times \sqrt{d\left(\log T + \frac{K^2 \log(K(\log T)/\rho)}{\rho^2}\right)} \sqrt{TPd \log\log d}. \tag{160}$$

$$\tag{161}$$

Here, in the last transformation, we used the standard Cauchy-Schwarz argument with $\sum_i N_i(T) = T$. The factor $Pd \log\log d$ is from the number of non-zero elements among $\{N_i(T)\}_i$; each phase uses an approximated G-optimal design with $O(d \log\log d)$ support, and thus there are $O(Pd \log\log d)$ non-zero elements of $N_i(T)$.

Therefore,

$$\mathbb{E}[\mathrm{Reg}(T)] \leq \mathbb{E}[\mathrm{Reg}(T)\mathbf{1}[\mathcal{G}]] + O(1) \tag{162}$$

$$= O\left(\sqrt{d\left(\log T + \frac{K^2 \log(K(\log T)/\rho)}{\rho^2}\right)} \sqrt{TPd \log\log d}\right) \tag{163}$$

$$\text{(by equation 117)} \tag{164}$$

$$= O\left(d\sqrt{T(\log T \log\log d)\left(\log T + \frac{K^2 \log(K(\log T)/\rho)}{\rho^2}\right)}\right), \tag{165}$$

$$\text{(by } P = O(\log T)) \tag{166}$$

which is the second regret bound of Theorem 16.

# I  Additional Simulations

## I.1  Tradeoff between regret and replicability

To discuss the tradeoff between regret and replicability, we conduct simulations on the same environment as Section 8 with different hyperparameters. Generally, there is a tradeoff between regret and replicability. However, the relation is not necessarily monotone, and thus we report the results for a wide range of hyperparameters. Smaller hyperparameters lead to smaller exploration and thus smaller regret.

The non-monotonicity of the $\rho_{\mathrm{emp}}$ can be explained by the following example: REC and RSE are phase-based algorithms. For example, assume that parameter $C = 0.5$ leads to 90%, 10% of runs to stop in phase 2 and 3, respectively. Assume $C = 1$ leads to 50%, 50% of runs to stop in phase 2 and 3, respectively. Assume $C = 2$ leads to 10%, 90% of runs to stop in phase 2 and 3, respectively. In this case, the nonreplicability is largest at $C = 1$ (nonreplicability of 0.5) because it equally splits the runs into two phases, while the nonreplicability is smaller at $C = 0.5$ and $C = 2$ because most runs stop in one phase.

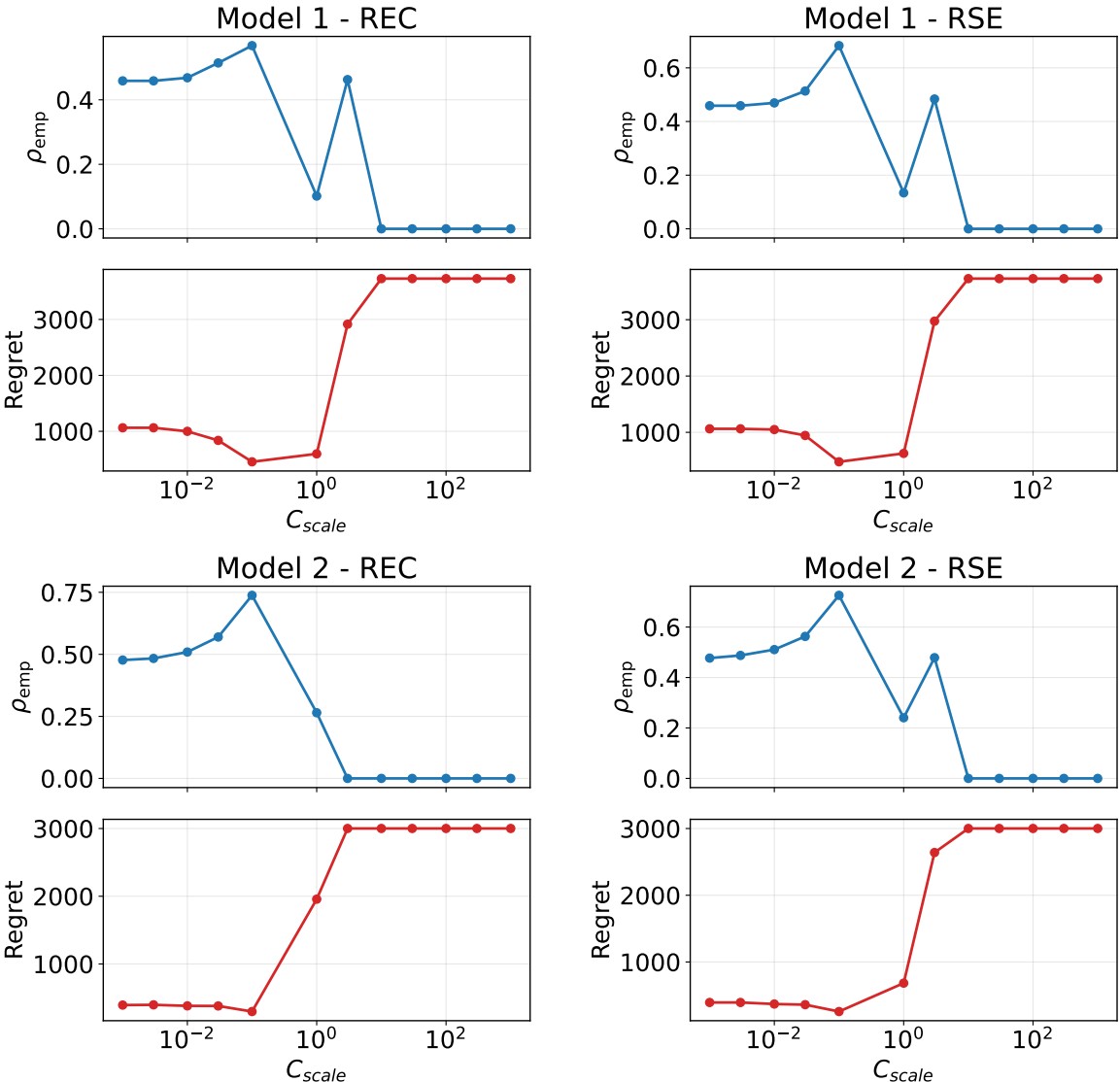

Figure 2: Replicability and Regret with different hyperparameters. The first row is for Model 1 and the second row is for Model 2. The left figure is for REC and the right figure is for RSE. $C_{\text{Scale}} = 1$ corresponds to the chosen parameters in the main paper.

### I.2 Scalability of algorithms

To discuss the scalability of algorithms, we conduct simulations on the same environment as Section 8 with different time horizons $T$. Table 4 summarizes the cumulative regret for tuned policies under different horizons $T$. These results show that regret of REC and RSE scales reasonably well with increasing $T$.

### I.3 Linear bandits

To discuss the advantage of linear representation (RLSE, Algorithm 3) over the non-linear one (RSE), we conduct simulations on linear bandits. We consider a bandit instance with $K = 5$, $d = 2$, and $\theta = (1, 1/2)$ where mean rewards $\mu_i = \boldsymbol{x}_i^\top \boldsymbol{\theta}$ are $(1.0, 0.7, 0.7, 0.7, 0.7)$ for $i = 1, 2, 3, 4, 5$. We set $\rho = 0.3$ and $T = 10^4$. The noise is generated from the standard Gaussian distribution. The results are shown in Figure 3. RLSE

Table 4: Cumulative regret for tuned policies under different horizons $T$.

| Model No. | T | RASMAB | REC | RSE | UCB1 |
|---|---|---|---|---|---|
| 1 | 1000 | 361.11 | 371.51 | 377.11 | 167.25 |
| 1 | 10000 | 1071.26 | 597.17 | 625.84 | 337.75 |
| 1 | 100000 | 1132.05 | 611.68 | 626.87 | 482.73 |
| 2 | 1000 | 119.47 | 399.15 | 160.17 | 40.62 |
| 2 | 10000 | 883.67 | 1956.87 | 683.10 | 114.39 |
| 2 | 100000 | 8858.16 | 2541.12 | 709.35 | 207.15 |

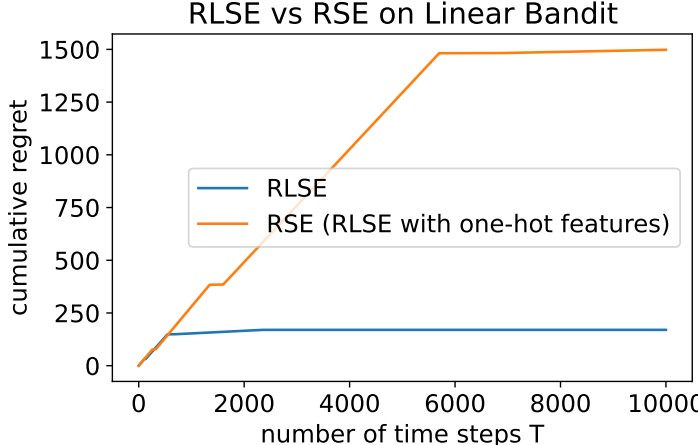

Figure 3: We compare the regret of RLSE and RSE in linear bandits. RLSE is Algorithm 3. RSE is equivalent to RLSE, except that it uses one-hot encoding. Because RSE does not leverage the linear structure, it suffers from higher regret than RLSE.

exploits the linear structure and thus achieves a smaller regret than RSE, which does not utilize the linear structure.

