# OpenReview forum: "Replicability is Asymptotically Free in Multi-armed Bandits"
_TMLR — Accepted by TMLR_

### Review · Reviewer_Jttg · 2026-03-17

**Summary Of Contributions:**

This paper studies replicable algorithms in multi-armed bandits. In addition to the regret-minimization objective in classic bandit algorithms such as UCB, the new replicability constraints require that the sequence of selected arms in two different runs be the same with high probability. Previous works show that to achieve $\rho$-replicability compared to non-replicable algorithms like UCB, one needs to pay a multiplicative factor of $K^2/\rho^2$. This paper proposes an explore-then-commit algorithm called REC and a successive elimination algorithm called RSE, both satisfies the $\rho$-replicability requirements. For both algorithms, the paper provides a regret bound demonstrating that replicability can be achieved with a universal constant multiplicative plus a lower-order term in $K^2 \log \log T/\rho^2$. The framework is extended to linear bandits, and some empirical results are shown.

**Additional Comments:**

The solid theoretical contribution of this paper should place itself above the bar of TMLR, but it is not publishable as it is concurrently. I recommend a major revision.

**Audience:**

Yes

**Audience Explanation:**

The paper studies the multi-armed bandit model and the linear bandit model, with a new replicability constraint. It is interesting for the bandit literature, and in particular, the constrained bandit community. It also provides an angle to justify the advantage of batched bandit algorithms, such as successive rejection versus fully adaptive algorithms such as UCB and TS. However, it seems the impact of this paper is largely limited to the theory community, where the main contribution mostly lies in better uncertainty quantification designs, such as tricks in designing better confidence bonuses. It is usually hard to convert this type of message to practical bandit applications.

**Claims And Evidence:**

Yes

**Claims Explanation:**

The main claim of the paper is that one could achieve replicability "almost for free" in the large $T$ regime. It is supported by the theorems, mainly Theorem 11. Compared to the classic regret bounds in the bandit literature, the RSE algorithm only incurs an additional $\log \log T$ term, which involves the replicability parameters. However, it is plausible that the leading $\log T$ term has a different universal constant compared to non-replicable algorithms, which the paper does not show. So "free" should be interpreted as a universal constant multiplicative plus lower-order terms in $T$.

**Requested Changes:**

Major Comments:

1. The intuition on why RSE achieves a much better regret bound compared to REC or (Esfandiari et al. 2023) is not clear from the discussion in the main text, unless reading the full technical proof. The authors should use some paragraphs explaining the main new designs of RSE and REC and how it leads to improved regret, and compare to (Esfandiari et al. 2023) if necessary.

2. The experiments are relatively weak and incomplete. As a constrained problem, the simulation section does not demonstrate whether the algorithms indeed satisfy the replicability requirement. A regret vs empirical nonreplication tradeoff curves, besides the only regret at a tuned target $\hat{\rho}$, should be shown. Linear bandit models should also be tested.

3. The empirical results of REC and RSE are still no match for UCB in Model 2, which is not consistent with the theory that "replicability is free". I suspect it's because the experiments are not run long enough with enough repetitions.

Minor comments in the proof:
1. The application of Theorem 9 is not formally correct as written for both the proofs of Theorems 10 and 11, because Lemma 18 / 22 bounds the global good event, but Theorem 9 already uses the union bound. Please fix this inconsistency by revising the statement.

2. The regret proof of Theorem 11 cites Lemma 22 under an event inconsistency, i.e., $G^{reg}$ seems to be different from $G$ in Lemma 22. Some bridging steps need to be stated.

3. For completeness, please include the proofs of Lemmas 23-25.

---

> ### Author Response · Authors · 2026-04-18
>
> We thank reviewer for their insightful comments and careful reading of the proof. Below, we answer the points raised.
>
> > The intuition on why RSE achieves a much better regret bound compared to REC or (Esfandiari et al. 2023) is not clear from the discussion in the main text, unless reading the full technical proof. The authors should use some paragraphs explaining the main new designs of RSE and REC and how it leads to improved regret, and compare to (Esfandiari et al. 2023) if necessary.
>
> While the analysis of RASMAB (Esfandiari et al. 2023) is suboptimal, under optimizer hyperparameters, it is similar to the individual arm elimination of RSE. We reorganized the explanation of simulation section (added Section 8.1) as follows: "While the regret analysis of RASMAB is based on a larger confidence bound than that of RSE, the empirical performance of RASMAB under the optimized hyperparameters is similar to RSE where the commitment (i.e., elimination of all arms simultaneously) is never performed.
> In other words, REC performs simultaneous elimination of all arms (commitment), whereas RASMAB eliminates each arm independently. RSE integrates both elimination mechanisms."
>
> > The experiments are relatively weak and incomplete. As a constrained problem, the simulation section does not demonstrate whether the algorithms indeed satisfy the replicability requirement. A regret vs empirical nonreplication tradeoff curves, besides the only regret at a tuned target $\hat{\rho}$, should be shown. Linear bandit models should also be tested.
>
> We optimize the hyperparameters (confidence widths) so that it satisfies the empirical probability of replication. We added Section I that is devoted to additional simulation.
>
> Section I.1 "Tradeoff between regret and replicability" shows how empirical $\rho_{emp}$ changes over the choice of hyperparameters. The relation is not perfectly monotone due to subtlety that we briefly discussed therein.
>
> Section I.3 "Linear bandits" test linear bandit and show that it is more efficient than RSE that does not use feature vectors.
>
> > The empirical results of REC and RSE are still no match for UCB in Model 2, which is not consistent with the theory that "replicability is free". I suspect it's because the experiments are not run long enough with enough repetitions.
>
> As you correctly noted in the previous part of review, the free replicability is to guarantee that the regret of these algorithms are *up to a constant factor* of UCB. Section I.2 "Scalability of algorithms" includes a new result with a larger value of $T=100000$ and it seems that the algorithm is scaling sublinearly even though it is several times worse to UCB.
>
> > The application of Theorem 9 is not formally correct as written for both the proofs of Theorems 10 and 11, because Lemma 18 / 22 bounds the global good event, but Theorem 9 already uses the union bound. Please fix this inconsistency by revising the statement.
>
> Thank you for pointing this out. We have revised Theorem 9 so that the first term now corresponds to the probability of the global good event.
>
> > The regret proof of Theorem 11 cites Lemma 22 under an event inconsistency, i.e., $G^{reg}$ seems to be different from $G$ in Lemma 22. Some bridging steps need to be stated.
>
> You are correct. Lemma 22 is for replicability whereas Theorem 11 is for regret analysis. These two use different events (that is why we can decompose the confidence level required for regret and confidence level to obtain the replicability with $o(\log T)$ cost) required for and should not be mixed. We corrected the proof of Theorem 11 and added a paragraph.
>
> > For completeness, please include the proofs of Lemmas 23-25.
>
> We have added the proofs and remarked that these proofs are similar to Lemma 19 and Lemma 20.
>
> We believe these suggestions have significantly improved the clarity and completeness of the paper.

---

### Review · Reviewer_4Dmd · 2026-03-25

**Summary Of Contributions:**

This work studies the design of a class of multi-arm bandit algorithms with replicability. Formally, with high probability over the noise in the rewards, a replicable algorithm should select the same sequence of arms. Two algorithms are provided: REC and RSE. The main idea is to divide the T rounds into P phases, and for each phase make sure that the same sequence of arms are selected. The proposed algorithms achieve a better regret bound compared to existing methods. and matches the optimal rate of regret when T is very large. The authors also propose the RLSE algorithm for the linear bandit problem. Finally, the authors complement their theoretical results with a simulation on some synthetic data.

**Audience:**

Yes

**Audience Explanation:**

I think the paper is in general very interesting. The paper is well written, and I have little difficulty in understanding the paper. Although I have some questions on the definition of replicability, which I will detail later, the paper did a great job in describing the problem setting and the context, comparing the new bounds with previous ones, and describing the intuition behind the proposed methods. The methods are also quite simple and can be quickly implemented in practice. I think people in the area of bandits will generally find this paper interesting.

**Claims And Evidence:**

Yes

**Claims Explanation:**

1. The proposed algorithms are very intuitive and reasonable
2. The bound makes sense to me. Although I didn't check the proof line by line, the bound looks generally correct compared to standard bounds
3. The authors also provide a lower bound for the regret, and compare it with the upper bound
4. The simulation provides empirical evidence that the proposed method can achieve lower regret than previous replicable algorithms, and the regret is close to that of UCB

**Requested Changes:**

My questions are mainly about the definition of "replicability", and the motivation behind the problem setting.
1. Definition 3 states that an algorithm is replicable only when it selects the exact same sequence of arms, with high probability over the noise in the rewards. Why is it so defined? Isn't this requirement too strong? I know that this definition comes from some previous papers, but I feel that it is unnecessarily strong for the purpose of experiment replicability. A weaker definition could be that each arm is selected for the same number of times. An even weaker one could be that the strongest arms are selected for the same number of times, since we might not even care about the weak arms after all. I am not sure why such a strong condition is chosen to be the definition of "replicability".
2. The consequence of this requirement being so strong is that the bound Eqn. (2) is loose when $\rho$ is small. For the $T$ typically used in practice, I feel that the dominant term in Eqn. (2) will be $\frac{K^2}{\rho^2}$. It will be much more ideal if a weaker notion of replicability allows the dependence on $\rho$ to be $\log \frac{1}{\rho}$. Another thing is Eqn. (6) is a stronger condition when $T$ is larger, which does not really make sense because normally doing more experiments should make the conclusion of the experiments more replicable.

Overall, I find this paper quite interesting. I lean towards accepting this paper if the above questions could be addressed.

---

> ### Author Response · Authors · 2026-04-18
>
> We thank the reviewer for their insightful comments. Below, we answer to the raised questions.
>
> > Definition 3 states that an algorithm is replicable only when it selects the exact same sequence of arms, with high probability over the noise in the rewards. Why is it so defined? Isn't this requirement too strong? I know that this definition comes from some previous papers, but I feel that it is unnecessarily strong for the purpose of experiment replicability. A weaker definition could be that each arm is selected for the same number of times. An even weaker one could be that the strongest arms are selected for the same number of times, since we might not even care about the weak arms after all. I am not sure why such a strong condition is chosen to be the definition of "replicability".
>
> Indeed, the current definition of replicability is important when each round is individual treatment: The definition makes sense when we consider the adaptive clinical trials as well as educational applications, where we want to ensure the treatment assignment of the same patient or the same student is the same across different runs. However, your proposed weaker notion is reasonable as a statistical testing viewpoint. We have added the discussion on your notion in "Discussion" Section 10.
>
> > The consequence of this requirement being so strong is that the bound Eqn. (2) is loose when $\rho$ is small. For the $T$ typically used in practice, I feel that the dominant term in Eqn. (2) will be $\frac{K^2}{\rho^2}$. It will be much more ideal if a weaker notion of replicability allows the dependence on $\rho$ to be $\log \frac{1}{\rho}$. Another thing is Eqn. (6) is a stronger condition when $T$ is larger, which does not really make sense because normally doing more experiments should make the conclusion of the experiments more replicable.
>
> Regarding whether the weaker notion of replicability (same number of draws but not necessarily same sequence), we consider it does not remove the necessity of the $1/(\rho \Delta)^2$ term. The term is *not* derived from the sequential nature, it is about estimating the gap $\Delta$ with a finer accuracy of $\rho \Delta$. Guarantee for the identical behavior is essentially discretization (algorithm draws arm $i$ for $1, 2^1, 2^2, 2^3,...,2^p$ times - we want to put our algorithm in the same bin $p$ across different estimate of $\hat{\Delta}$). If $\hat{\Delta}$ fluctuate with $O(\hat{\Delta})$ magnitude, we cannot guarantee $1-\rho$ fraction lies in the same discretization bin.
>
> There is a rescue provided in the literature of known gap case introduced by Reviewer HxbW. If we know the magnitude of $\Delta$ beforehand, we hypothesize we can remove $1/(\rho \Delta)^2$ term. This looks exciting because we can use knowledge of previous data to ensure a very efficient design of experiment.
>
> We think the main point of the even weaker replicability (only guarantee the number of draw of the best arm) is to improve the dependence on $K$. We are not sure but it might remove $K^2$ dependence for RSE.

---

> > ### Comment · Reviewer_4Dmd · 2026-05-10
> >
> > I thank the authors for the response. I am satisifed with the rebuttal and have no more question at this moment.

---

### Review · Reviewer_HxbW · 2026-04-04

**Summary Of Contributions:**

This paper studies replicability in stochastic bandits and makes a nice point: the cost of replicability does not have to multiply the usual regret term, and for large $T$ it can become a lower-order additive term. The phased framework is clean, and REC/RSE are natural algorithms under this perspective. I also like that the paper includes both a general upper-bound framework and a lower bound for the two-armed case.

**Audience:**

Yes

**Audience Explanation:**

Researchers in the bandit community may find this topic interesting.

**Broader Impact Concerns:**

I do not have major broader-impact concerns.

**Claims And Evidence:**

Yes

**Claims Explanation:**

All claims are correct and rigorously proved.

**Requested Changes:**

**Questions**:

Q1. For REC, the key design choice is the switching rule from exploration to commitment, namely committing once $\hat\Delta(p)$ exceeds a randomized confidence threshold. This clearly drives both regret and replicability. I would like to understand better what is lost if this switching time is made more aggressive. Is the main issue worse regret, failure of the $\rho$-replicability guarantee, or both? Right now the paper shows that the chosen rule works, but it is less clear how tight or principled this stage-switching choice is. Regarding the literature on explore-then-commit, several highly relevant works should be discussed carefully, including:

[1] Aurélien Garivier, Tor Lattimore, and Emilie Kaufmann. On Explore-Then-Commit Strategies. In Advances in Neural Information Processing Systems 29 (NeurIPS 2016), 2016.
[2] Aurélien Garivier, Pierre Ménard, and Gilles Stoltz. Explore First, Exploit Next: The True Shape of Regret in Bandit Problems. Mathematics of Operations Research, 44(2):377–399, 2019.
[3] Tianyuan Jin, Pan Xu, Xiaokui Xiao, and Quanquan Gu. Double Explore-then-Commit: Asymptotic Optimality and Beyond. In Proceedings of the 34th Conference on Learning Theory (COLT 2021), Proceedings of Machine Learning Research 134:2584–2633, 2021.
[4] Tianyuan Jin, Jing Tang, Pan Xu, Keke Huang, Xiaokui Xiao, and Quanquan Gu. Almost Optimal Anytime Algorithm for Batched Multi-Armed Bandits. In Proceedings of the 38th International Conference on Machine Learning (ICML 2021), Proceedings of Machine Learning Research 139:5065–5073, 2021.

---

Q2. I am less convinced by the application story than by the theory. The formal notion here is very specific: reproducing the same action sequence with probability at least $1-\rho$. But the experiments are still standard regret simulations, and do not really demonstrate the practical benefit that motivates the paper, such as more reliable statistical testing or more stable scientific conclusions across reruns.

---

Q3. The lower bound is interesting, but it is only for the two-armed Bernoulli setting. Since one of the main messages is that the proposed regret is essentially optimal, I would have liked a bit more discussion of how much this lower bound should be taken as evidence for the general $K$-armed case.

---

Q4. The empirical section is useful, but still fairly limited. It mainly uses two synthetic models, and the methods are tuned to an empirical nonreplication level. This is enough to illustrate the idea, but not enough to fully understand robustness in practice.

---

> ### Author Response · Authors · 2026-04-18
>
> Dear Reviewer HxbW,
>
> We appreciate your insightful comments. Below we answer to your individual questions
>
> > Q1. For REC, the key design choice is the switching rule from exploration to commitment, namely committing once $\hat\Delta(p)$
>  exceeds a randomized confidence threshold. This clearly drives both regret and replicability. I would like to understand better what is lost if this switching time is made more aggressive. Is the main issue worse regret, failure of the
> $\rho$-replicability guarantee, or both? Right now the paper shows that the chosen rule works, but it is less clear how tight or principled this stage-switching choice is.
>
> Indeed, our contribution is that the confidence level for regret and the confidence level for replicability is *independent*. To minimize regret, we must estimate the gap $\Delta$ with confidence level $1/T$ (Lai and Robbins 1985), which leads to $\log T/\Delta^2$ bound. To ensure replicability, we must estimate this gap with *a finer accuracy* of $\rho \Delta$ but does not need a confidence level of $1/T$, which leads to $1/(\rho \hat\Delta)^2$ bound but it is $o(\log T)$. REC's confidence bound is chosen to satisfy both.
>
> > Regarding the literature on explore-then-commit, several highly relevant works should be discussed carefully, including:
>
> We added them in the related work section. Indeed, the known-gap case [1,4] seems interesting. The cost of $1/(\rho \Delta)^2$ exploration comes from the fact that we need to estimate the gap with a finer accuracy, and if we have a good estimate of gap, we hypothesize we can remove this term, which is exciting. This is related to Reviewer 4Dmd's comment. Thank you for introducing relevant articles.
>
> > Q2. I am less convinced by the application story than by the theory. The formal notion here is very specific: reproducing the same action sequence with probability at least $1-\rho$. But the experiments are still standard regret simulations, and do not really demonstrate the practical benefit that motivates the paper, such as more reliable statistical testing or more stable scientific conclusions across reruns.
>
> In the discussion section, We added the following paragraph: The notion of replicability in this paper is defined to be the probability of two independent runs of an algorithm resulting in the same sequence of decisions. The definition makes sense when we consider the adaptive clinical trials as well as educational applications, where we want to ensure the treatment assignment of the same patient or the same student is the same across different runs.
>
> > Q3. The lower bound is interesting, but it is only for the two-armed Bernoulli setting. Since one of the main messages is that the proposed regret is essentially optimal, I would have liked a bit more discussion of how much this lower bound should be taken as evidence for the general $K$-armed case.
>
> The point of nontriviality has been discussed in Remark 2 "Extension for $K$-armed Lower Bound," but we are happy to elaborate if needed.
>
> > Q4. The empirical section is useful, but still fairly limited. It mainly uses two synthetic models, and the methods are tuned to an empirical nonreplication level. This is enough to illustrate the idea, but not enough to fully understand robustness in practice.
>
> In response to Reviewer Jttg's request, we have added an additional experiment section that measures the scalability of algorithm in $T$ as well as robustness check of $\rho_{emp}$ over the sweep of hyperparameters. We also added linear bandit simulation. That said, we agree that this paper is mainly theoretical.

---

> > ### Comment · Reviewer_HxbW · 2026-05-07
> >
> > Thank you for the response and the revision. I appreciate the authors' efforts to address my comments.
> >
> > The clarification on the switching threshold is helpful, and the added discussion of ETC/batched bandits and additional simulations improve the paper. My remaining concerns are minor. The application motivation is still somewhat limited, and the lower bound is still only for the two-armed Bernoulli case, so I encourage the authors to state this limitation more explicitly.
> >
> > Overall, I think the revision addresses my main concerns sufficiently, and I am satisfied with the paper.

---

### Decision · Action_Editor_Yr24 · 2026-05-19

**Recommendation:** Accept as is

**Additional Comments:**

To further strengthen the paper, the authors could benefit from a more detailed discussion on the practical implications of the asymptotic result. Specifically, a deeper analysis of the finite-time trade-offs, where the cost of replicability is most pronounced, would enhance the paper's practical relevance.

**Audience:**

Yes

**Audience Explanation:**

The work is highly theoretical and provides a significant conceptual advance in the field of bandit algorithms. The paper successfully establishes a new boundary condition for the trade-off between replicability and regret. This would be of interest to many audience of TMLR who are studying bandit and reinforcement learning algorithms.

**Claims And Evidence:**

Yes

**Claims Explanation:**

This paper introduces a principled approach to replicable stochastic multi-armed bandit algorithms, addressing the critical need for reproducibility in published findings. The core contribution is the demonstration that the additional regret cost typically associated with ensuring replicability is asymptotically unnecessary when the time horizon is sufficiently large, provided the confidence bounds are chosen carefully.

---

> ### Author Response · Authors · 2026-06-03
> **Camera ready submission**
>
> Dear Action Editor,
>
> We have submitted the camera-ready revision. In response to your comment, we added a new subsection, “10.2 Summary for Practitioners.” This section discusses: (1) the implication, namely that standard UCB and TS are non-replicable, whereas explore-then-commit and elimination algorithms can be replicable. Moreover, it discusses (2) the finite-time trade-offs, on the range of $T$ where the cost of replicability arises. The discussion here is intended to be practical rather than highly detailed, which we believe is reasonable given that the decision is “accept as is.”
>
> Best,
>
> Authors